# Filling the gap between topological insulator nanomaterials and triboelectric nanogenerators

Mengjiao Li[1,2,3,8], Hong-Wei Lu[1,8], Shu-Wei Wang[1,4], Rei-Ping Li[5], Jiann-Yeu Chen[6,7], Wen-Shuo Chuang[5], Feng-Shou Yang[2], Yen-Fu Lin [2,6✉], Chih-Yen Chen[5✉] & Ying-Chih Lai [1,6,7✉]

Reliable energy modules and higher-sensitivity, higher-density, lower-powered sensing systems are constantly required to develop wearable electronics and the Internet of Things technology. As an emerging technology, triboelectric nanogenerators have been potentially guiding the landscape of sustainable power units and energy-efficient sensors. However, the existing triboelectric series is primarily populated by polymers and rubbers, limiting triboelectric sensing plasticity to some extent owing to their stiff surface electronic structures. To enrich the current triboelectric group, we explore the triboelectric properties of the topological insulator nanofilm by Kelvin probe force microscopy and reveal its relatively positive electrification charging performance. Both the larger surface potential difference and the conductive surface states of the nanofilms synergistically improve the charge transfer behavior between the selected triboelectric media, endowing the topological insulator-based triboelectric nanogenerator with considerable output performance. Besides serving as a wearable power source, the ultra-compact device array demonstrates innovative system-level sensing capabilities, including precise monitoring of dynamic objects and real-time signal control at the human-machine interface. This work fills the blank between topological quantum matters and triboelectric nanogenerators and, more importantly, exploits the significant potential of topological insulator nanofilms for self-powered flexible/wearable electronics and scalable sensing technologies.

---

[1] Department of Materials Science and Engineering, National Chung Hsing University, Taichung 40227, Taiwan. [2] Department of Physics, National Chung Hsing University, Taichung 40227, Taiwan. [3] Ming Hsieh Department of Electrical and Computer Engineering, University of Southern California, Los Angeles, CA 90089, USA. [4] Francis Bitter Magnet Lab, Massachusetts Institute of Technology, Cambridge, MA 02139, USA. [5] Department of Materials and Optoelectronic Science, National Sun Yat-Sen University, Kaohsiung 80424, Taiwan. [6] i-Center for Advanced Science and Technology, National Chung Hsing University, Taichung 40227, Taiwan. [7] Innovation and Development Center of Sustainable Agriculture, National Chung Hsing University, Taichung 40227, Taiwan. [8] These authors contributed equally: Mengjiao Li, Hong-Wei Lu. ✉email: yenfulin@nchu.edu.tw; cychen@mail.nsysu.edu.tw; yclai@nchu.edu.tw

The development of wearable electronics and the Internet of Things (IoT) strongly depends on the advance of both reliable power sources and sensing systems[1–4]. Although various power modules—relying on piezoelectric, pyroelectric, photoelectric, and electrochemical effects, as well as environmentally renewable energy—have gained significant momentum, their cost-effective application remains challenging given issues including lower outputs, chemical stability, and colossal volume[5–7]. Moreover, the conversion efficiency of thermal, optical, mechanical, and clean energy is largely limited owing to rigid operating conditions or stimuli direction dependence. These drawbacks of current energy harvesting systems highlight the necessity of exploring sustainable energy generators: triboelectric nanogenerators (TENGs)[8–10]. TENGs have emerged as a conjunction of tribology and interfacial charge transfer, building an unprecedented network to effectively harvest the mechanical energy distributed around our daily life[11,12]. Friction-induced charge separation and transfer on the surface of triboelectric materials enable TENGs to detect and recognize external fluctuations by recording electric signals. This capability, combined with several further advantages, including easy fabrication, optional working mode, and multi-directional force adaptability, paves the way for producing self-sufficient triboelectric sensors and triboelectronics[13–19].

Improving the performance of TENGs requires enhancement of the triboelectric charge densities that are strongly associated with the electron affinity difference between selected triboelectric materials and surface-contact modification engineering[20,21]. On the one hand, enlarging the effective contact area in terms of dimensions and morphology is a direct strategy. A spectrum of techniques, including hydrothermal synthesis, templating fabrication, ink-jet printing, and plasma treatments, have been used for this purpose. Typical examples include using Au nanoparticles to decorate the triboelectric film, knitting triboelectric networks with silk-nanofibers, and fabricating vertical $TiO_2$ nanoflakes on Ti foils[8,22–24]. On the other hand, introducing elaborate charge generation dynamics, such as charge trapping processes, modified dielectric constants, and tailored electronic structures, also contribute to enhancing the triboelectric characteristics. It has been shown that incorporating $BaTiO_3$ nanoparticles, 2D reduced-graphene oxides, or $MoS_2$ nanoflakes into organic dielectric films can modify the dielectric constant and charge trapping dynamics as a result of tunneling effects[25–28]. Although promising advances are being made in nano modification engineering, joining both strategies into a single medium to achieve boosted triboelectric dynamics without compromising the surface plasticity remains challenging owing to the difficulties in shrinking the hybrid systems. In addition, most triboelectric materials are governed by polymers or rubbers, essentially hampering the possibilities for minimizing or functionalizing the active layer at the atomic limit to adapt for scaling down in future electronics[21,29–31]. Therefore, alternative triboelectric materials that are fundamentally different in terms of physical properties and electronic structures are urgently required to achieve energy-efficient, reliable, technologically simple, and scalable sensory systems for wearable electronics.

Topological insulators (TIs) are emerging electronic phases with gapped bulk bands and gapless surface states that show tremendous potential for applications in optoelectronics, quantum computing, and spintronics[32–35]. The surface conductions of TIs become prominent when their thicknesses are reduced to the nanoscale, which makes TIs vital candidates for future nanoelectronics[36,37]. Bismuth telluride ($Bi_2Te_3$), a compound that has been extensively studied for its thermoelectric properties, was found to be a 3D TI with a quintuple-layered structure and gained significant attention in recent years[38,39]. $Bi_2Te_3$ nanofilms

are usually prepared by mechanical exfoliation or molecular beam epitaxy (MBE), which yield samples with excellent quality yet low scalability. By contrast, chemical solvothermal synthesis is considered more efficient, by which the $Bi_2Te_3$ nanofilms are formed by many $Bi_2Te_3$ nanoplates ($Bi_2Te_3$ NPs) and can be made in a large area using the spin-coating method. This solution-based method provides good control over the shape and size of $Bi_2Te_3$ NPs, affording a viable platform for exploring the practical applications of $Bi_2Te_3$[40–42]. For example, the combination of high surface-volume ratio, topological surface conduction, and unique dielectric behavior has endowed $Bi_2Te_3$ NPs with significant potential for use in fast logic transistors, efficient thermoelectric catalysis, wide-band photodetectors, and microwave absorbers[43–46]. Albeit never involving the field of nanogenerators, TIs' unique surface conducting properties make them ideal candidates for TENGs since the triboelectrification is strongly dominated by the surface charge transfer process between tribolayers[47,48]. Therefore, a significant effort in exploring the triboelectric characteristics of TIs is required to fill the research gap between topological materials and triboelectric energy devices.

In this work, to explore the potential of TIs in energy harvesters and energy-efficient electronics, we investigate the electrical performance of a TI-enabled triboelectric nanogenerator (TI-TENG) by assembling $Bi_2Te_3$ NPs on a flexible substrate. Its triboelectric charging ability is evaluated using Kelvin probe force microscopy (KPFM) analysis and found to lie between nylon and Al in the existing triboelectric series. The surface conducting property improved contact behavior, and larger surface potential difference with Kapton endows $Bi_2Te_3$-based TI-TENGs with enhanced triboelectric charge transferability and considerable output power performance. TI-TENGs equipping with external capacitors enable to serve as flexible power sources to drive portable electronics. In addition, TI-TENG sensors exhibit precise sensing performance with small device spacings, which can realize the construction of an ultra-compact sensory system to implement object monitoring, real-time signal processing, and self-powered human–machine interfacial applications (music players and game controllers). These findings introduce TI nanomaterials into the triboelectric series and serve as a significant paradigm for functional materials in the fields of power devices and energy-efficient electronics.

## Results

**The triboelectric polarity of $Bi_2Te_3$-NP films**. Considering the advantages of the solution-based method over fabricating high-yield and low-cost nanofilms, in this work, a chemical solvothermal synthesis method was used to explore the triboelectric behavior of TI triboelectric layers[40,49]. Figure 1a shows schematic diagrams of the growth of $Bi_2Te_3$ NPs, indicating that the formation procedure begins with dissolved Bi and Te ions in solution. After heating, Te and Bi ions are first reduced to Te and Bi atoms (Fig. 1a(i)); then numerous Te atoms aggregate to form Te nanorods owing to their unique helical-chain crystal structures (Fig. 1a(ii))[50]. The assembled Te nanorods then provide available sites for the heterogeneous nucleation and alloying of $Bi_2Te_3$ nanoparticles. Increasing the reaction time facilitates the emergence of hexagonal nanoplates (Fig. 1a(iii)). To maintain the electronic properties of the assembled TI films, ethylene glycol was used as a surfactant to ensure a gentle electron transfer between nanoplates. Details can be found in the Methods.

Figure 1b–h shows the characterization of the fabricated $Bi_2Te_3$ NPs in terms of morphology, phase, composition, and uniformity. Atomic force microscopy (AFM) and scanning electron microscopy (SEM) analysis showed that the thickness of the $Bi_2Te_3$ NPs

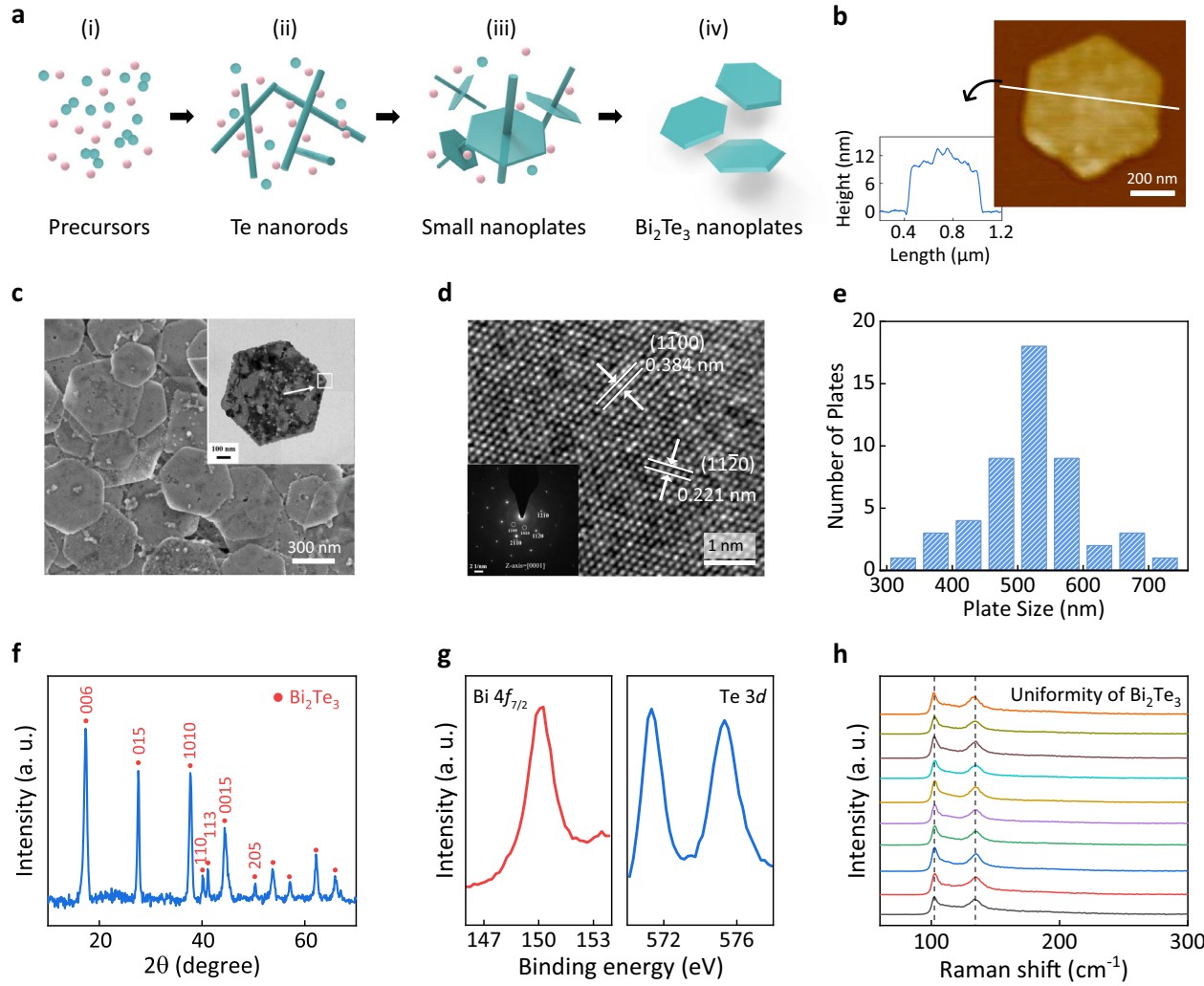

**Fig. 1 Materials characterization of Bi$_2$Te$_3$ NPs. a** Schematic illustration of growth mechanism for Bi$_2$Te$_3$ NPs. **b** AFM image of a Bi$_2$Te$_3$ NP and the corresponding height/width profile along the solid line across the nanoplate. **c** SEM and TEM images (inset) of the hexagonal Bi$_2$Te$_3$ NPs. **d** HRTEM image of Bi$_2$Te$_3$ sample and the corresponding SAED image with bright hexagonally symmetric spots. **e** Statistic of the size distribution of Bi$_2$Te$_3$ NPs by AFM analysis. **f** XRD pattern of the fabricated Bi$_2$Te$_3$ NPs. **g** High-resolution XPS spectra of Bi 4$f$ and Te 3$d$ region, respectively. **h** Raman spectra were collected from stochastic sites on an individual Bi$_2$Te$_3$ NP. Source data are provided as a Source Data file.

was around 10 nm and the average width between two parallel sides was 500–550 nm (Fig. 1c, e). In the high-resolution transmission electron microscopy image (Fig. 1d), the well-resolved lattice spacings of 0.22 and 0.38 nm can be indexed as the (11$\bar{2}$0) and (1$\bar{1}$00) planes, respectively, indicating the Bi$_2$Te$_3$ NPs have high crystallinity[51]. In addition to distinct diffraction spots, weak diffraction rings in the inset are attributed to the stacked NPs in various axes. The phase purity of Bi$_2$Te$_3$ NPs was examined by X-ray diffraction (Fig. 1f), and only featured peaks of the rhombohedral structure were detected (JCPDS #15-0863)[52]. High-resolution X-ray photoelectron spectra (XPS) for the regions of Bi and Te (Fig. 1g) confirmed both the composition and the valence states of the Bi$_2$Te$_3$ NPs. In addition, 10 Raman spectra that exhibited consistent $E_g$ and $A_{1g}$ phonon vibration modes were randomly collected from a single nanoplate, revealing the high homogeneity of the nanoplates synthesized using the wet chemical route and thereby the high uniformity of the Bi$_2$Te$_3$-NP films for assembling TI-TENGs (Fig. 1h).

To explore the triboelectric performance of TI nanomaterials, a vertical-mode TI-TENG was fabricated for electrical characterization. Figure 2a shows the device structure. It consists of Bi$_2$Te$_3$ film, Kapton, and indium tin oxide (ITO), serving as two

triboelectric layers and electrodes, respectively. The schematic diagrams of the crystal structure and energy band of Bi$_2$Te$_3$ in Fig. 2b(i) highlight its building block—quintuple layer (QL) and the unique surface conductive features. Each QL cell consists of five layers, which are stacked by a sequence of Te(1)–Bi–Te(2)–Bi–Te(1) along the z-direction and terminated by a Te(1) layer at both ends. Compared with the strong interaction within each QL cell, the van der Waals force between adjacent QL cells is much weaker, leading to the preferential cleave surface of Te atomic layer[38,49,53,54]. The preparation procedures are described in detail in the supporting information (Supplementary Fig. 1). SEM images in Fig. 2b show two different surface morphologies for the Bi$_2$Te$_3$ film assembled by NPs (Bi$_2$Te$_3$-NP film), revealing that its compactness and regularity closely depend on the coating dosage of Bi$_2$Te$_3$ NP colloid. Figure 2c presents the one-cycle voltage outputs of TI-TENG and corresponding working mechanism schematics based on the triboelectrification and electrostatic induction effects[55]. In the first stage, the work function difference between the two tribolayers leads to charge transfer from the Bi$_2$Te$_3$-NP film to Kapton when making contact. When they are brought away from each other (second stage), electrons flow from the bottom

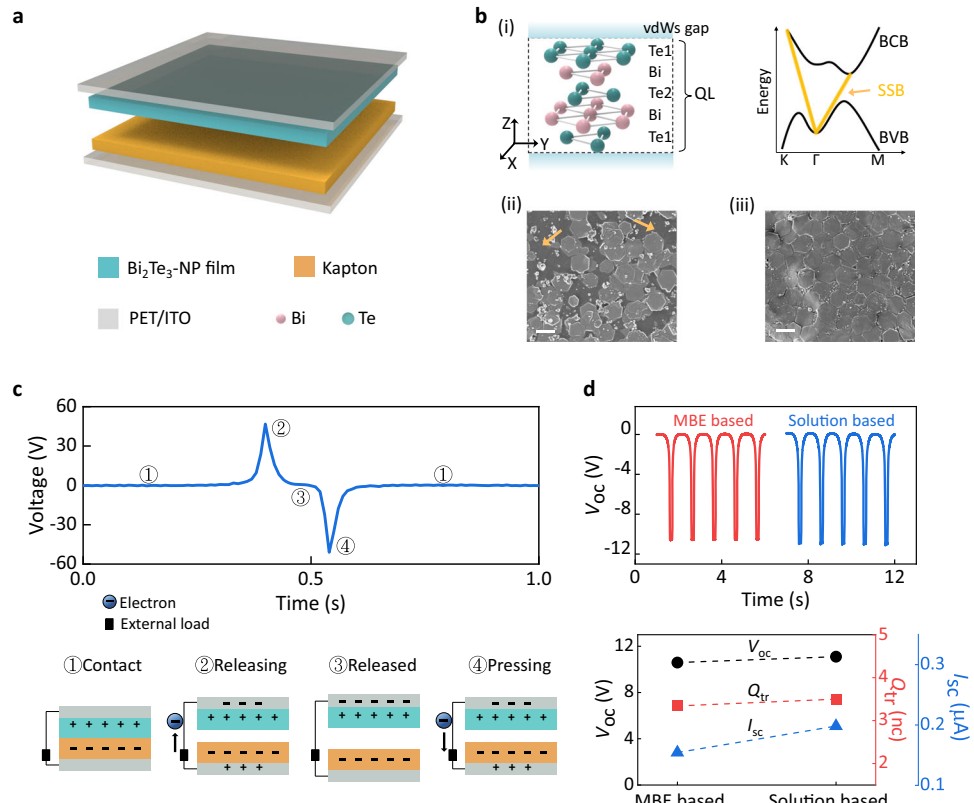

**Fig. 2 The demonstration of the TI-TENG. a** Structural schematic diagram of a TI-TENG based on $Bi_2Te_3$-NP film. **b** Crystal structure (i) of the quintuple layer (QL) $Bi_2Te_3$ (Te(1)–Bi–Te(2)–Bi–Te(1)) and a sketch of the band structure diagram with typical bulk bandgap (bulk conductive band/BCB, bulk valance band/BVB, black lines) and unique conductive surface state band (yellow line, SSB). The vdWs gap represents the van der Waals gap between each QL. SEM images of the TI films with different coating doses of $Bi_2Te_3$ NP colloid ((ii) 2 mL and (iii) 10 mL). The scale bars are 500 nm and the arrows highlight the discontinuity of $Bi_2Te_3$-NP film with a lower coating dose. **c** Output voltage signal of solution-based TI-TENG (8 mL, 5 × 5 cm²) in one cycle. The down panels show the working principle for several different stages. **d** Comparison plots of the output properties of TI-TENGs assembled with solution-based $Bi_2Te_3$-NP film (8 mL, 0.5 cm²) and MBE-based $Bi_2Te_3$ film, which reveal their slight differences. Source data are provided as a Source Data file.

electrode (Kapton-side) to the top electrode ($Bi_2Te_3$-NP film-side) to screen the triboelectric charges, resulting in a positive voltage signal. Electron motion completes at the third stage (fully released), quenching the output. When the separation distance is reduced, the number of the induced charges on two electrodes decreases, resulting in reverse electron motion as well as a reverse output in the external circuits (fourth stage)[56]. Such outputs obtained from the paired $Bi_2Te_3$-NP film and Kapton indicate a more positive triboelectric charging ability for $Bi_2Te_3$-NP film than Kapton, which agrees with the simulated results (Supplementary Fig. 2).

The triboelectric performance of the TIs was further validated by investigating the output behavior of the MBE-grown TI films. The crystallinity was examined by Raman spectroscopy (Supplementary Fig. 3). For a fair comparison, the measurement conditions for MBE TI-TENG were made as similar as possible to the solution-based TI-TENG. As shown in Fig. 2d and Supplementary Figs. 4 and 5, the two TENGs with different preparation methods delivered the same triboelectric polarity and similar output curves, indicating the low dependence of the triboelectric charging characteristics of TI films on the synthetic approach. Notably, the slight performance difference between these two devices could originate from multiple factors, such as the different effective contact areas or interlayer interactions[57].

The accurate positioning of a triboelectric medium in the triboelectric series significantly determines its availability and applicability. To investigate the triboelectric order of $Bi_2Te_3$-NP film, several TENGs consisting of various $Bi_2Te_3$-triboelectric medium pairs were prepared, including nylon, aluminum (Al), paper, polymethyl methacrylate (PMMA), copper (Cu), polydimethylsiloxane (PDMS), fluorinated ethylene propylene (FEP), and polyvinyl chloride (PVC) (Supplementary Fig. 6). All of the collected voltage signals in Fig. 3a show the same electrical polarity under the releasing/pressing operations, except for the $Bi_2Te_3$-nylon pair. Given the positive polarity of nylon in the triboelectric series, it was deduced that $Bi_2Te_3$ lies to the right of nylon, where the triboelectric polarity is defined by colored arrows and the left arrow points to more positive polarity, and the right arrow points to more negative polarity[20,21]. The uniform voltage polarity from Al to PVC suggests their more negative charging abilities compared with $Bi_2Te_3$. Therefore, $Bi_2Te_3$ is empirically expected to lie between nylon and Al, showing a relatively positive electrification behavior. Such a clear positioning for the $Bi_2Te_3$-NP film also suggests its narrow selectivity for positive partners and broad selectivity for negative partners, providing a reasonable direction for further exploration and application of Ti-based TENGs.

It is known that the charge transfer behavior between two triboelectric layers determines their triboelectric polarities and the generated voltage signals. The surface charge transfer process strongly depends on the effective work functions and potential difference of the contacted media, thus necessitating the examination of the surface potential of various materials by KPFM measurements to confirm the obtained triboelectric orders. Figure 3b shows the relative potential differences ($\Delta E = E_{tip} - E_{sam}$) as a function of typical media, where $E_{tip}$ and $E_{sam}$ represent the

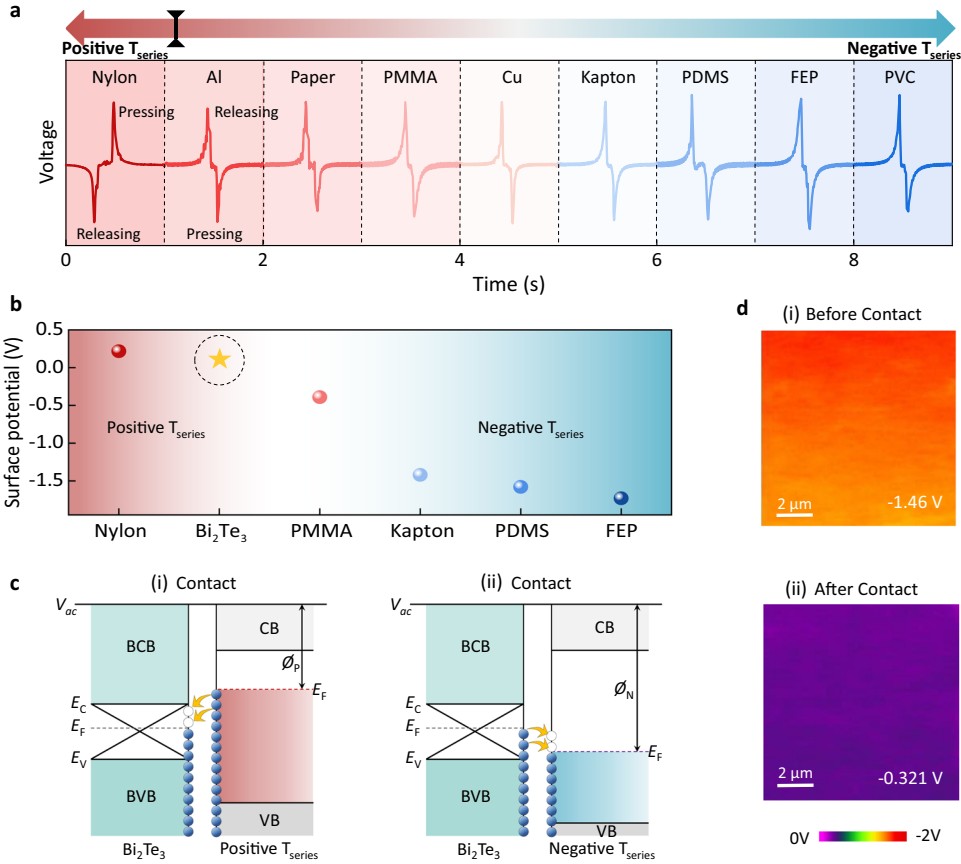

**Fig. 3 Triboelectric series positioning of the TI films. a** Output signals of the solution-based TI-TENGs with various media as the counter triboelectric layer, which hint at the triboelectric series of TI film between nylon and Al film. Note that the red arrow indicates positively triboelectric ($T_{series}$) charging properties and the blue arrow indicates negatively triboelectric charging properties. The black symbol on the arrow proposes the possible position of TI materials. **b** The measured surface potential of various triboelectric layers using the KPFM technique. The star symbol and the dashed circle denote the $Bi_2Te_3$-NP film. **c** Energy band diagram for $Bi_2Te_3$-NP film when contacting (i) positively electrified materials and (ii) negatively electrified materials. The crossed symbols indicate the conductive surface state of TI material. $E_F$ is the Fermi energy and $E_{c(v)}$ is the bulk conductive (valence) band edge of $Bi_2Te_3$. $\emptyset_P$ and $\emptyset_n$ are the work functions for positive and negative triboelectric materials, respectively. **d** Surface potential variation of Kapton (i) before and (ii) after contact with $Bi_2Te_3$-NP film. Source data are provided as a Source Data file.

surface potentials for the probe tip and measured sample, respectively. Nylon showing the largest $\Delta E$ (0.2 V) indicates it has the smallest work function potential among all characterized samples. Thus, the surface electron transfer from nylon to $Bi_2Te_3$ is expected when they contact to equalize the Fermi level difference, as illustrated in the energy band diagram (Fig. 3c(i)). This results in positive charge electrification on the surface of nylon and negative charge electrification on the surface of $Bi_2Te_3$-NP film, defining their triboelectric polarities.

By contrast, $Bi_2Te_3$ exhibits higher $\Delta E$ (0.1 V) than the others (below 0 V). This leads to the reverse charge exchange process and positively electrified TIs, which coincide closely with the electrical measurements in Fig. 3a and confirm the triboelectric order of $Bi_2Te_3$. In addition, the dynamic potential variation of contact triboelectric layers was characterized to validate the surface charge transfer mechanisms. As shown in Fig. 3d(i) (taking the TI-Kapton pair as an example), before contact, the Kapton delivers a larger $|\Delta E|$ due to its larger work function than that of the probe tip. After contact with TI, electron transfer from TI to Kapton, resulting in the elevated Fermi level, subsequently decreased the potential difference (Fig. 3d(ii)). The uniform surface potential variation of Kapton is indicative of the electron transport analysis in Fig. 3c. Such electrification effect of TIs was further validated by examining the triboelectric behavior of another typical TI material —$Bi_2Se_3$. Under the same measurement conditions, $Bi_2Se_3$-NP

film-based TENG exhibits similar triboelectric performance with $Bi_2Te_3$ TI-TENG (Supplementary Fig. 7). The systematical investigation of the surface morphology and surface potential variation further indicates that the triboelectric performance of TI-TENGs originates from the synergistic contributions from surface potential, contact behavior, and the conducting property of TI materials. Considering that the charge transfer process is closely associated with the value of $\Delta E$, Kapton was selected as the counterpart triboelectric medium in the subsequent experiments because there is a relatively large potential difference between $Bi_2Te_3$-NP film and Kapton[58,59]. In addition, Kapton has been frequently selected as the triboelectric material in previous studies on TENGs with new material or new structures (Supplementary Fig. 8). Thus, Kapton is used as the counterpart material in our work to make a fair comparison with other materials without compromising the output performance.

**The triboelectric performance of $Bi_2Te_3$-NP TENGs.** Understanding the triboelectric properties of TI films motivated us to systematically investigate the output performance of TI-TENGs. The investigation was conducted on several important factors: the contact area, operating frequency, external load, and endurance. First, various TI-Kapton pairs with different amounts of $Bi_2Te_3$ NP colloid were prepared to characterize their energy harvesting capability. As shown in Fig. 4a, both the open-circuit voltage

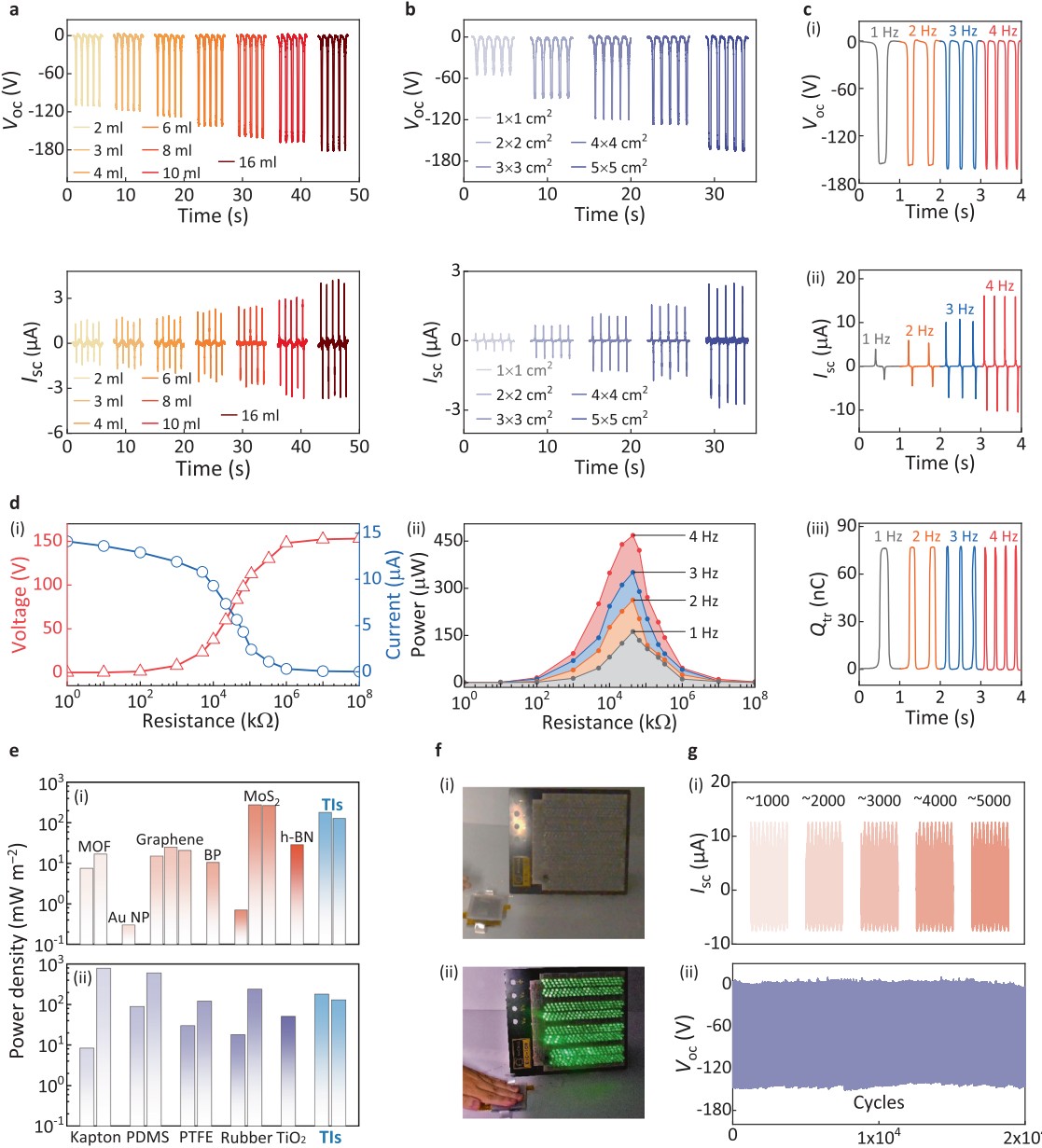

**Fig. 4 Electric performance characterizations of the TI-TENG. a** Output voltage and current signals of solution-based TI-TENGs with various coating doses of $Bi_2Te_3$ NP colloid from 2 to 16 mL. **b** Output voltage and current signals of the solution-based TI-TENGs with various areas from $1 \times 1$ to $5 \times 5$ cm². **c** Output properties of the solution-based TI-TENG (8 mL, $5 \times 5$ cm²) depending on different operating frequencies. **d** Output voltage and current signals (i) of the solution-based TI-TENG as a function of various resistances, also the corresponding output power (ii) under different measurement frequencies. **e** Comparison plots of the output power density among previously reported TENGs based on (i) emerging nanomaterials, (ii) conventional materials, and TIs ($Bi_2Te_3$ and $Bi_2Se_3$). **f** Optical image of the (i) in-series LEDs and (ii) LEDs lit by a solution-based TI-TENG. **g** Endurance test of the TI-TENG under the continuous operation of (i) 5000 and (ii) 20,000 cycles. Source data are provided as a Source Data file.

($V_{oc}$) and short-current ($I_{sc}$) vary distinctly depending on the doses of $Bi_2Te_3$ NPs. It was observed that even a small dose (2 mL) of $Bi_2Te_3$ NPs enabled a high $V_{oc}$ above 100 V, showing a better performance of TI-TENGs. The output voltage reached 180 V when the dose of $Bi_2Te_3$ NPs was increased to 16 mL; the slow increase rate could be ascribed to the gradual saturation of the surface coverage ratio of the $Bi_2Te_3$-NP film. Contributions from the polyethylene terephthalate (PET) substrate were excluded by examining the output properties of PET-Kapton paired-TENG without TI film (Supplementary Fig. 9). The importance of the contact area of the triboelectric media was further investigated. Note that the dose of $Bi_2Te_3$ NPs colloid was proportional to the area of the films for a fair comparison. In Fig. 4b, the

output voltage shows a proportional increase with the increasing area of the TI film, which is consistent with the proportional relationship between the total amount of the transferred charges and the contact area[47,58]. The fitting result is provided in Supplementary Fig. 10. The tunability of the output—via both the dose of $Bi_2Te_3$ NPs and the size of the $Bi_2Te_3$-NP films—suggests the high practicality of TI-TENGs.

Furthermore, the electrical output properties of TI-TENGs with a $5 \times 5$-cm² and 8-mL dose TI-film were explored under various measurement conditions. As expected, in Fig. 4c(i) and (ii), both $V_{oc}$ and $Q_{tr}$ (transferred charges) show steady outputs up to almost 160 V and 80 nC with increasing operating frequency from 1 to 4 Hz. While, the short contact time at high frequency leads to a fast

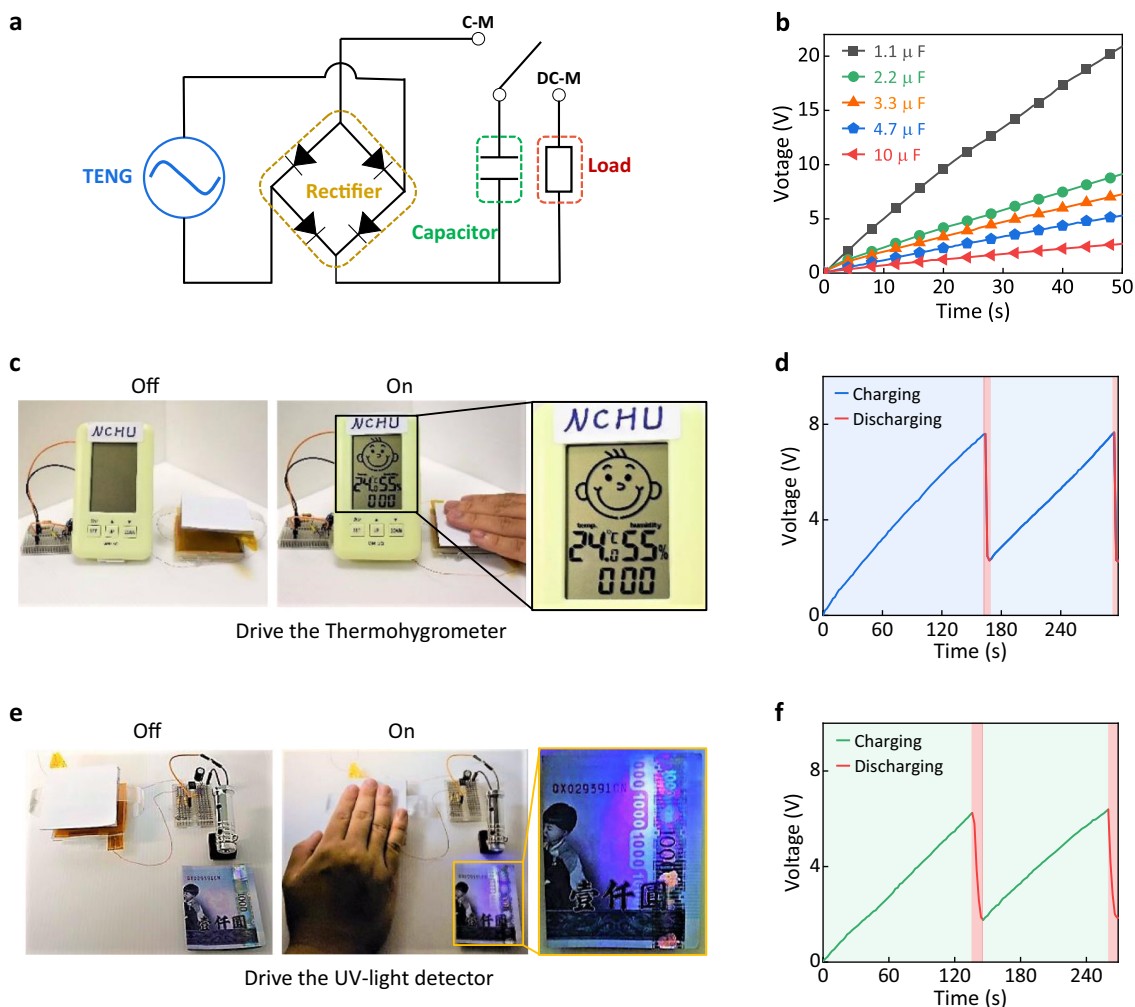

**Fig. 5 Demonstration of the TI-TENG power sources. a** Equivalent circuit of the TI-TENG-based electric power system by equipping with capacitance and external loads. Note that C-M and DC-M represent the charging and discharging working mode, respectively. **b** Output voltage signals of the solution-based TI-TENG depending on the equipped capacitance under the charging mode. Schematic illustration of the designed electric power system for driving commercial loads under the discharging working mode, such as driving **c** a thermohygrometer and **e** a UV-light detector. **d**, **f** The corresponding charging and discharging profiles for two cycles. Source data are provided as a Source Data file.

charge flow, responsible for increasing $I_{sc}$ from 5 to 15 μA (Fig. 4c(ii)). We further investigated the performance of TI-TENGs as power sources to drive loads by introducing various resistances into the external circuit. As shown in Fig. 4d(i), both the extracted voltage and current peaks change slightly under a small resistance, while both start a steep variation at 30 MΩ and symmetrically enter into the maximum and minimum states, respectively. Correspondingly, the variation in output power as a function of external resistance is expected to exhibit a hump shape at 30 MΩ. They also present uniform tendency under different operating frequencies, which is indicative of the reliability of the TI-TENG as a power source. In Fig. 4e(i), the output power density of TI-TENGs could reach 180 mW m$^{-2}$, which could be competitive with emerging nanomaterials-based power sources, including MoS$_2$-based, graphene-based, and metal oxide frameworks-based counterparts. While the plot in Fig. 4e(ii) eclipses the performance advantage of TI-TENGs compared with many conventional TENGs, especially polymer-based devices, potentially directing the improvement target for TI-TENGs[8,22,60–74].

The powering performance of a 5 × 5-cm$^2$ TENG was visually demonstrated by driving external loads. As illustrated in Fig. 4f (and Supplementary Movie 1), 480 light-emitting diodes in series were simultaneously lit up thanks to the high outputs of TI-TENG. It is

important to assess the durability and reproducibility of the power systems. As shown in Fig. 4g, the output current and voltage of the TI-TENG remained stable after thousands of cycles, demonstrating excellent endurance performance. Note that the negligible fluctuation over 20000 cycles could be attributed to the gentle material transfer between two triboelectric layers or the noise signal from the external circuits, which is consistent with the surface morphology variation of Kapton (Supplementary Fig. 11).

**Power sources based on Bi$_2$Te$_3$-NP TENGs.** In general, TENGs instantaneously produce high output signals during the dynamic external operations, which likely limits their direct use in conditions that require consistent power sources[9,75]. Unsatisfactory utilization makes TENGs uncompetitive in diverse power source markets. As a result, a mini-type electric generation-power system has been developed to improve the efficiency of TENGs by collecting, transforming, and storing mechanical energy as electric energy for optional uses. Figure 5a shows the power management circuit of the TI-TENG, which consists of a capacitor, a bridge, and a load. Working states are switchable between the TENG and load, which correspond to the charging and discharging modes, respectively. To demonstrate the broad applicability of TI-TENGs, capacitors with different capacitances were employed in the charging mode, ranging

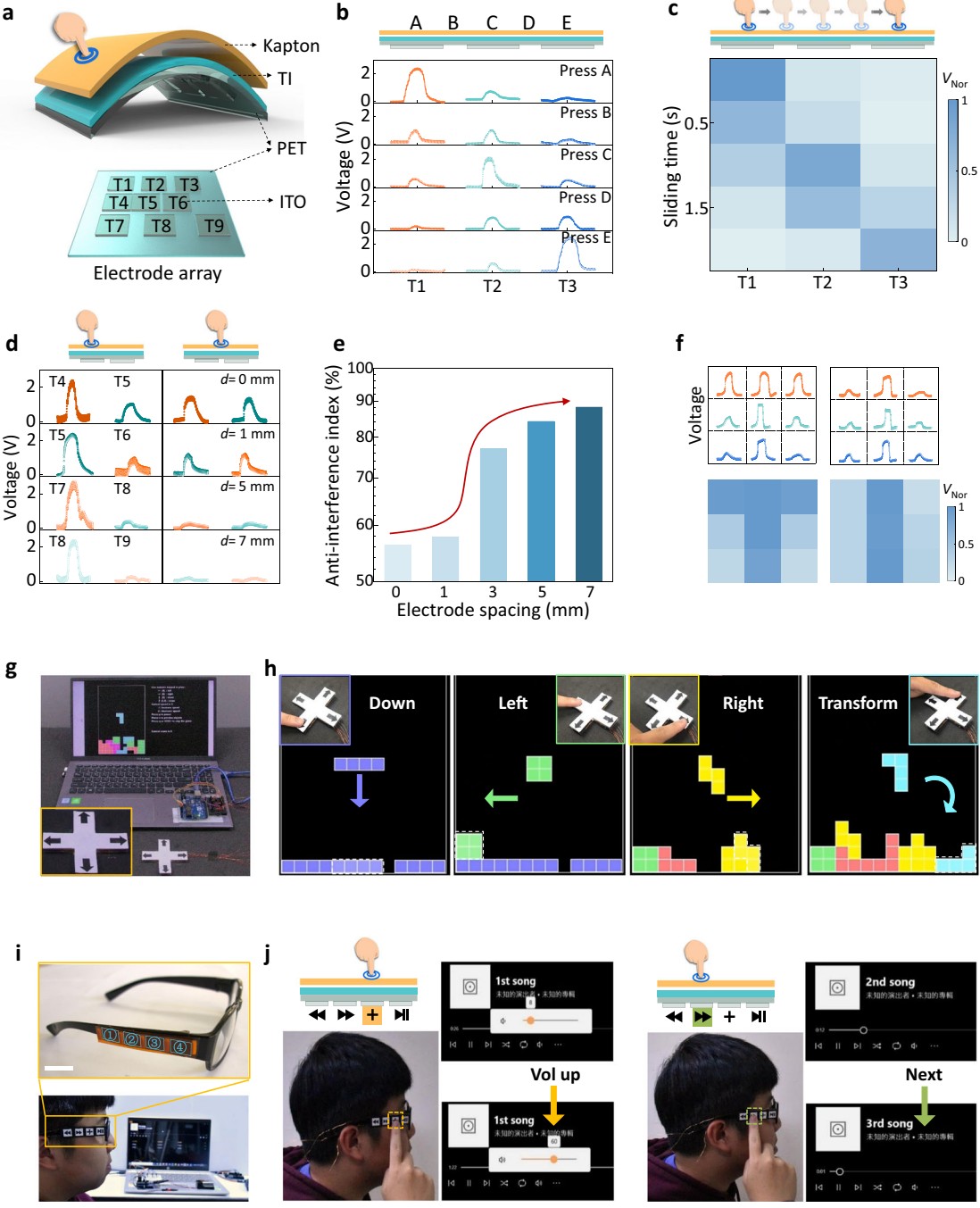

**Fig. 6 TI-TENG self-powered sensors and system-level applications. a** Structural schematics of the designed TI-TENG sensor array with various spacings. The bottom view shows the relative distribution of nine ITO electrodes, corresponding to nine independent TENGs (numbering from T1 to T9). **b** Output voltage signals of TI–T3 by touching different sites. The top panel illustrates the corresponding pressing sites of A–E. **c** Output voltage signals mapped in electrode positions and measurement time when sliding across the surface of TI film from T1 to T3 (sliding speed: 13 mm s$^{-1}$). $V_{Nor}$ means the normalized voltage signals. **d** Output signals of T4–T9 by pressing on the top of the electrodes of T4, T5, T7, T8 (left panel), and middle sites (right panel). For a fair comparison, the tapping sites highlighted at the top panel were operated with a uniform contact force and area. **e** Plots of the statistical anti-interference index of TI-TENG sensory system depending on different spacings from panel (d), which shows the anti-interference abilities of tactile sensors at each spacing to improve the system-level design. Note that the index was extracted from the ratio of $(V_0 - V)/V_0$, where $V$ and $V_0$ represent the measured voltage value for each cell with possible disturbance and without disturbance, respectively. A curve arrow is used to guide eyes. **f** Recorded output voltage curves and normalized maps of T1–T9 when touching with different objects of letter T and I. **g** Image of a game controller integrated with four TI-TENG sensors and zoom-in sensory system (inset). **h** Images of various demonstrations of TI-TENG sensors-enabled operating instructions, including Down, Left, Right, and Transform. **i** Images of a pair of smart glasses-enabled wireless music players. The upper panel illustrates four integrated TI-TENG sensors on the side bracket. TENG cells with numbers 1–4 represent directions of Last, Next, Vol up (voice up), and Play/Stop, respectively. **j** Demonstration of operating the Vol up and Next instructions via triggering self-powered TI-TENG sensors. Scale bar in (**g**) and (**i**): 2 cm. Source data are provided as a Source Data file.

from 1.1 to 10 µF. During the charging mode (C-M), the evolution of voltages in Fig. 5b shows distinct charging behaviors depending on the connected capacitors. The fastest increase rate was observed for the 1.1-µF capacitor and the slowest increase rate was observed for the 10-µF capacitor, obeying the inversely proportional relationship between the capacitances and charging voltage. Various loads were then connected to examine the power ability of the TI-TENG power system at the discharging mode (DC-M). Figure 5c–f illustrates the practical charging-discharging curves of different loads. After charging for ~130 s, the collected energy was capable of driving a commercial thermohygrometer and a UV-light detector (Supplementary Movie 2 and Movie 3), which resulted in a synchronous decline of the voltage signals. Continuously alternating curves indicate the reproducibility of the charging-discharging process and the sustainability of TI-TENG power systems for an energy-efficient lifestyle.

**Self-powered sensors based on $Bi_2Te_3$-NP TENGs.** Based on their excellent output performance, the demonstrations of TI-TENGs in tactile sensing and system-level human–device interface applications were systematically explored. Figure 6a and Supplementary Fig. 12 show an integrated TI-TENG sensory system consisting of 9 individual cells (numbered from T1 to T9) created by patterning discrete ITO electrode arrays with various spacings ($d = 0, 1, 3, 5, 7$ mm). Different touching sites on Kapton correspond to distinguishable output signals owing to different amounts of induced charge. In Fig. 6b, touching directly above the ITO electrode (site A, C, and E) of T1, T2, and T3 ($d = 3$ mm) gives three identical outputs with weak by-products at either side, indicating its excellent resolution under a spacing of 3 mm. While, it is difficult to maintain this anti-interference behavior at the middle sites B and D ($d = 1.5$ mm), leading to two similar outputs from two adjacent TENGs. To improve the integrability of sensory systems, the device resolution was further examined by designing device spacings from 0 to 7 mm (Fig. 6d). The dependence of the extracted variation of the sensing resolution on spacing $d$ (Fig. 6e) is believed to direct the system-level uses of TI-TENG sensors with operating validity.

Output mapping as a function of time and device position shows the dynamic sensing capability of the TI-TENG sensory array (Fig. 6c). Under a uniform sliding speed across the surface of T1–T3, the collected curves for each cell with distinct fluctuations endowed them with precise tactile perception of dynamic objects. The TI-TENG sensory system demonstrated further applicability in self-powered human–machine interfaces. A $3 \times 3$ array with an identical spacing of 3 mm was designed to monitor touched objects (Supplementary Fig. 13), akin to the smart electronic skin of an artificial hand. Figure 6f shows the obtained voltage signals that were proportional to the contact areas between $Bi_2Te_3$-NP film and Kapton without significant noise. Thus, the shape of the monitored objects, letters T and I, were clearly distinguished according to the voltage mapping (bottom panel of Fig. 6f and Supplementary Fig. 13).

In addition, the excellent controllability and integrability of the TI-TENG sensory system allowed their application in self-powered human-machine interfaces. Figure 6g shows an integrated system that consists of four TI-TENG sensors presented as a game controller. By coordinating the circuit design, pressing each operating key enables the launch of the TENG sensor below, thereby executing corresponding instructions (Supplementary Movie 4). A pair of smart glasses was further demonstrated as a wireless wearable controller by integrating miniature TI-TENG sensors. Lightly touching the second or the third sensors freely controlled the instructions of Vol up or Next (Fig. 6j and

Supplementary Movie 5). Such demonstrations of self-powered microcontrollers suggest promising uses for the integrated TI-TENG sensing systems in the fields of human–machine interfaces, supporting further research aimed at smart robotics.

## Discussion

In summary, a TI—a unique triboelectric medium with favorable surface charge properties—was introduced into the triboelectric series and the field of triboelectric energy. Systematic KPFM analysis combined with electronic transport behavior investigation revealed the triboelectric charging characteristics of the TI film, directing the rational design of the TI-based TENGs. Benefiting from the enhanced surface charge transfer process, the Kapton-$Bi_2Te_3$ film paired TI-TENG exhibited considerable output power, reliable energy-harvesting capabilities, and the ability to drive portable electronics. Furthermore, TI-TENGs as self-powered sensors demonstrated anti-interference sensing resolution, enabling the construction of ultra-compact sensory systems for human-machine interfacial applications, including a game controller and a wireless smart music player. We, therefore, believe that engineering TI nanomaterials to extend the triboelectric series will bridge the fields of topological quantum matters and wearable/smart electronics toward diverse functionalities.

## Methods

**Synthesis of $Bi_2Te_3$ nanoplates.** Bismuth nitrate pentahydrate ($Bi(NO_3)_3 \cdot 5H_2O$, 99.999%, Acros Organi, 0.2 mmol), sodium tellurite ($Na_2TeO_3$, 99.5%, ALFA, UK, 0.3 mmol), polyvinyl pyrrolidone (PVP, M.W. 40000, ALFA, UK, 2 mmol), and sodium hydroxide (NaOH, 97%, SHOWA) were dissolved in 10 mL of ethylene glycol. To examine the effect of alkaline solution, different amounts of NaOH (2–5 mmol) were used. The mixture was then heated under reflux in a three-neck flask. The temperature of the mixture was maintained at 190 ℃, respectively, for 3-h periods, and then cooled to room temperature. The synthesized sample (theoretically, 0.1 mmol $Bi_2Te_3$) was centrifuged at 6700 g for 8 min with a solvent mixture of 5 mL of acetone and 10 mL of isopropanol. The precipitates were dispersed in 5 mL of acetone and 10 mL of isopropanol and cleaned using an ultrasonicator for three times. The percent yield was around 75% considering the unavoidable loss during material collection and cleaning. The final products were dispersed into isopropyl alcohol solution (10 mmol $L^{-1}$) for further device fabrication and dropped onto the Si substrate using the spin coating method for further characterization.

**Characterization.** For materials characterizations, AFM (Bruker Dimension Icon), X-ray diffraction (XRD, D8, Cu-Kα radiation (λ = 1.54 Å), scanning rate of 0.0125º $s^{-1}$), XPS (ULVAC-PHI, PHI 5000 VersaProbe), Raman scattering spectra (LabRam HR-800, Horiba Jobin Yvon, the wavelength of laser: 488 nm), scanning electron microscope (SEM, JEOL JSM-6330), and field-emission transmission electron microscope (TEM, Tecnai G2, acceleration voltage of 120 kV) were employed to investigate the morphology, components, valence states, crystal structure, and crystallinity of the as-synthesized $Bi_2Te_3$ nanoplates. Kelvin Probe Force Microscope (KPFM, Bruker Dimension Icon) measurements were further conducted in the tapping mode with a conductive probe to characterize the relative surface potential of various triboelectric media. Typical electrical measurements of the TI-TENGs and TI-TENG sensor arrays were implemented on a commercial linear mechanical motor with controlling programs. Keithley electrometer system (Keithley Instruments, Cleveland) was used to record all the output signals of TI-TENGs.

## Date availability

Source Data are provided with this paper. All the experimental results of the main manuscript and Supplementary Information are available on google drive via the accession code https://drive.google.com/drive/folders/1O2_7Zg27x0TAiW9zxuPePwD-RLhRMu46?usp=sharing. Source data are provided with this paper.

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

## Acknowledgements

This work is supported by the Ministry of Science and Technology (109-2112-M-005-012-MY3;110-2221-E-005-038; 110-2224-E-005-001; and 110-2112-M-110-017-MY2) in Taiwan and "Innovation and Development Center of Sustainable Agriculture" from The Featured Areas Research Center Program within the framework of the Higher Education Sprout Project by the Ministry of Education (MOE) in Taiwan.

## Author contributions

M.J.L. and Y.-C.L. conceived and designed the experiments and led the research. H.-W.L. contributed to the device fabrication and electrical measurements. M.J.L., H.-W.L., and Y.-C.L. analyzed the data. R.-P.L., W.-S.C., and C.-Y.C. contributed to the synthesis and characterizations of nanomaterials. J.-Y.C. and F.-S.Y. performed the Kelvin Probe Force Microscope measurements. S.-W.W. contributed knowledge about topological insulators. M.J.L., H.-W.L., and Y.-C.L. prepared and revised the manuscript. Y.-F.L., C.-Y.C., and Y.-C.L. supervised the research. All the authors discussed the results and commented on the paper.

## Competing interests

The authors declare no competing interests.
