## [Peer Review File · Nature Communications]

REVIEWER COMMENTS

Reviewer #1 (Remarks to the Author):

This manuscript presents the topological insulator nanofilm-based triboelectric nanogenerators (TI-TENG) by assembling the 2D bismuth telluride (Bi_2Te_3) NPs on a flexible substrate. Previous nanomodification-engineered TENGs have struggled to further improve the efficiency and plasticity due to the limited materials including polymers and rubbers. The author proposed the potential of TI nanoplates for TENG by analyzing the surface potential with many counter layers and comparing the effects of coating dose and dimension on power density. It is remarkable that this TI-TENG first utilized the typical 2D TI materials to achieve not only a down-scaling and efficiency-improving, which enabled the self-powered real-time human-machine interface applications. However, the reviewer thinks this manuscript needs additional data supplement to be published in Nature Communications after major revision. The required revision data are summarized as follows.

1. Regarding the relative potential differences (ΔE) (Fig. 3b, d), the author mentioned that the efficiency of charge transfer is related with ΔE . The detailed explanation about the selection of Kapton is recommended for clear understanding. The reviewer suggest that the relationship equation should be included for the authorship because nylon and FEP shows the largest and smallest ΔE , respectively. (Kapton does belong to neither side.)

2. In addition, the experimental data should be added for explaining the merits of 2D Bi_2Te_3 nanoplates. For example, Fig. 4e includes the power density of only nanomaterials. As illustrated in introduction, the figures should describe what the 2D TI materials improved compared to the conventional TENGs.

3. The quality and configuration of some figures should be improved for better understanding. So below papers are recommended to be referred to enhance the informativeness of figures.

A. "Achieving high-resolution pressure mapping via flexible GaN/ZnO nanowire LEDs array by piezo-phototronic effect." Nano Energy, 58, 633-640 (2019)

B. "Monolithic flexible vertical GaN light-emitting diodes for a transparent wireless brain optical stimulator." Advanced Materials, 30, 1800649 (2018)

Reviewer #2 (Remarks to the Author):

In this manuscript, the authors investigated the feasibility to use low-cost Bi_2Te_3 -NP as the platform for energy-efficient triboelectric applications. Through detailed electric performance characterizations and various system demonstrations of the TI-based TENG, the authors did "fill the gap" and unfold a new avenue for the TI materials-based applications. In general, the work

presented in this manuscript is quite original and inspiring in the TI research field, and it deserves the publication once the authors can address the following comments.

1. The biggest missing puzzle in this manuscript is what the role of the topological surface states in enhancing the performance of TI-based TENG. According to the authors' argument, "the triboelectric charge densities that are strongly associated with the electron affinity difference between selected triboelectric materials and surface-contact modification engineering". If this is true, it seems that the only key factor of TI for TENG application is the relatively large surface potential (Fig. 3b). Under such circumstances, I do not recommend not to overstate the topological surface properties in this manuscript. After all, Bi₂Te₃ has been widely used long before it was "re-named" as topological insulators.
2. Likewise, the authors should clearly elaborate the enhancement of TENG performance of Bi₂Te₃-NP is due to the increased surface-to-bulk volume or topological surface state features (e.g., higher surface mobility, spin-to-momentum locking, and suppressed back-scattering). It should be noted that the surface state contribution is normally negligible at room temperature owing to the dominant thermal-activated bulk condition.
3. In Fig. 2, the authors also compared the TI-NP with MBE-grown TI film. However, there is no information regarding the MBE-based TI control samples, e.g., film thickness, area, substrate, carrier density etc.
4. Line 74 – Bi₂Te₃ is conventionally categorized as a 3D TI instead of 2D TIs (e.g., CdTe/HgTe, InAs/GaSb heterostructures).
5. The band structure of Bi₂Te₃ in Fig. 2b is not correct.
6. The authors claimed that "the electrical performance of the Bi₂Te₃-NP film in Figure 2d was slightly better than that of MBE-based Bi₂Te₃, which is attributed to the increased surface electronic states in the assembled-nanoplate film as a result of abundant defective edges." If this is true, I seem that the high performance of TI-based TENG is more related to the surface defects other than the original topological surface states.
7. Besides, I notice that the band diagrams shown in Fig. 3c do not include the TI surface states.

Reviewer #3 (Remarks to the Author):

This paper describes an experimental work using a topological insulator material, Bi₂Te₃, to construct triboelectric nanogenerators. As authors claim, triboelectric nanogenerators are becoming a good choice in powering self-powered devices, sensors, etc. Although there are innumerable works and substantial development of the practical approaches in making up these devices, since the devices are made up mostly of organic polymers (at least on one electrode), plasticity is an issue. The new class of materials the authors suggest and the example they show in their report seems to solve some of the problems in current TENG technology.

The authors clearly show the preparation of the material and the manufacturing of the devices with the new material. They provide two ways of materials preparation (solution-based and MBE-based), both of which work to deliver the targeted material - with slight differences in the device. Also, working with conventional materials on the other electrode, they built a set of TENG devices and have shown that the new material added to the device increases the charge transfer abilities considerably. They also described the charge generation mechanism by showing that the charge generation is due to an electron transfer between the surfaces – the direction of which is dependent on the ranking of the materials with respect to each other in the triboelectric series. This mechanism is supported well with the KPFM potential maps.

In addition to all these, the display of applications with the new material shows versatility.

Overall, the claims are supported well and the methodology used meets the standards in the field.

I suggest only minor additions to the manuscript:

1) It could have been interesting to show ‘identical material’ contact with the Bi₂Te₃ material on both electrodes. I guess there would be no charging, which I would expect due to the electron transfer mechanism. In cases with organic polymers, this is never the case.

2) The mechanism and in prolonged operation in real-life cases, there can be material transfer from the other electrode material (polymer) to the topological insulator, which I think is mechanically harder. Then the contact may be a polymer(deposited on Bi₂Te₃) – polymer rather than the pristine material contact. Can authors comment on this? Is it possible to use no polymers with the new material TENGs?

3) It was mentioned in the text that ‘The tunability of the output—via both the dose of Bi₂Te₃ NPs and the size of the Bi₂Te₃-NP films—enable good scalability of TI-TENGs for practical uses.’

Was there some correlation between the surface area (size) of the nanoplates and the performance metrics of the device /efficiency of charge transfer?

4) Does the synthesis provide quintuple layers all the time? For the unfamiliar reader, it is better to describe this part a bit.

Reviewers' Comment Letter

Response to Reviewer #1:

Reviewer #1: This manuscript presents the topological insulator nanofilm-based triboelectric nanogenerators (TI-TENG) by assembling the 2D bismuth telluride (Bi_2Te_3) NPs on a flexible substrate. Previous nanomodification-engineered TENGs have struggled to further improve the efficiency and plasticity due to the limited materials including polymers and rubbers. The author proposed the potential of TI nanoplates for TENG by analyzing the surface potential with many counter layers and comparing the effects of coating dose and dimension on power density. It is remarkable that this TI-TENG first utilized the typical 2D TI materials to achieve not only a down-scaling and efficiency-improving, which enabled the self-powered real-time human-machine interface applications. However, the reviewer thinks this manuscript needs additional data supplement to be published in Nature Communications after major revision. The required revision data are summarized as follows.

We thank Reviewer #1 sincerely for carefully reviewing our manuscript and raising the professional comments that are significant for further enhancing our work and improving the quality of the manuscript. In this work, we successively conducted a series of experiments, including nanomaterial synthesis and characterizations, device fabrication, and KPFM analysis, to explore the triboelectric properties of a new material—TI. We further systematically performed electrical characterizations and system-level applications of TI-TENGs in self-powered sensors and human-machine interfaces. As requested by Reviewer #1, more detailed data (**1. A collection of the Kapton-based TI-TENGs; 2. The comparison of the output performance between conventional TENGs and TI-TENGs; 3. The endurance test of TI-TENG for prolonged operations**) has been supplemented to emphasize the significance of the TI-TENGs and further enrich our manuscript. Also, careful modifications have been made to improve the quality of the figures (including **Figure 2, Figure 3, and Figure 4**). To make a clear explanation, the reviewer's comments and suggestions have been carefully considered and replied point-by-point. The relative corrections are highlighted in the revised version of the manuscript.

Comment 1:

Regarding the relative potential differences (ΔE) (Fig. 3b, d), the author mentioned that the efficiency of charge transfer is related with ΔE . The detailed explanation about the selection of Kapton is recommended for clear understanding. The reviewer suggest that the relationship equation should be included for the authorship because nylon and FEP shows the largest and smallest ΔE , respectively. (Kapton does belong to neither side)

Response 1:

The authors would like to thank Reviewer #1 for pointing out the professional comment about choosing Kapton as the triboelectric layer in our work, and we agree with the reviewer's viewpoint.

In this work, to check the triboelectric order of TI materials in the triboelectric series, we first selected a series of typical triboelectric materials as the references. After qualitatively determining the triboelectric ranking of Bi₂Te₃-NP films between Nylon and Al, we employed Kapton-based TI-TENGs for further investigations. The reasons for choosing Kapton as the counter-triboelectric material can be categorized into two aspects:

1. Extensive research of TENGs has revealed that the output performance of TENGs is closely related to the transferred charge densities. Quantitative studies (please refer to *Nano Lett.* 2013, 13, 2771; *Nat. Commun.* 2019, 10, 1427; *Nano Lett.* 2014, 14, 1567) further showed that a large work function difference and potential difference direct the selection of triboelectric material. This is because the charge transfer process between triboelectric materials can be expressed as $\frac{d\sigma}{dn} = kV_c - pV_e\sigma$, where σ , V_c , V_e , n , k , and p are the accumulated surface charge density (unit: C/m²), work function difference between two contacting materials, image charge potential induced by existing charges, the number of friction cycles, charge efficiency coefficient, and charging impedance coefficient, respectively. The KPFM results in Figure 3 (main text) reveal that the relatively larger potential difference between Bi₂Te₃-NP film and Kapton can benefit the output performance, albeit slightly lower than FEP.
2. Kapton is a frequently-used triboelectric material in recent TENGs studies, particularly for TENGs exploiting new materials or new working modes. As shown in **Figure R1**, Kapton has been used as the triboelectric layer in the metal-organic framework (MOF)-based TENG, MoS₂-based TENG, reduced graphene oxide (rGO)-based TENG, carbon nanotube/Mxene-based TENG, alternating mode TENG, and rotational TENG. Thus, to make a fair comparison with previous works, we considered choosing Kapton as the counter layer to perform the following experiments.

Therefore, inspired by Reviewer #1's suggestions, to make a detailed explanation about the material selection, the relative sentence "Notably, the number of transferred charges between triboelectric layers is closely associated with the value of ΔE , thus Kapton was selected as the counterpart triboelectric medium in the following investigation to achieve high outputs of TI-TENGs." has been revised as "Considering that the charge transfer process closely associates with the value of ΔE , Kapton was selected as the counterpart triboelectric medium in the subsequent experiments because there is a relatively large potential difference between Bi₂Te₃-NP film and Kapton^{57, 58}. In addition, Kapton has been frequently selected as the triboelectric material in previous studies on TENGs with new material or new structures (**Figure S8**). Thus, Kapton is used as the counterpart material in our work to make a fair comparison with other new materials without compromising the output performance." on **Page 9**. The collected comparison plot of Kapton-based TI-TENGs in **Figure R1** is supplemented in the revised manuscript as **Figure S8**. Also, two relative references are added as **Ref. 57** (*Nano Lett.* 2013, 13, 2771) and **Ref. 58** (*Nano Lett.* 2014, 14, 1567) to support our discussions. Thanks so much for Reviewer #1's constructive comment on why Kapton is the

triboelectric layer, and we hope that our explanation can offer a clear answer to Reviewer #1.

Year	Structure of TENGs	Reference
2017	Al/Kapton/MoS ₂ -Al/PET	ACS Nano 2017, 11, 8356
2017	Al/Kapton/rGO-Al	Nano Energy 2017, 32, 542
2018	Al/Kapton-Al-Kapton/Al	Nano Energy 2018, 51, 721
2019	PET/Cu/Kapton-MOF/PET/ITO (Metal-Organic Framework)	Adv. Energy Mater. 2019, 1803581
2020	Al/Kapton-MOF/Al	Adv. Funct. Mater. 2020, 1910162
2020	Kapton-based ternary electrification layered TENG	ACS Nano 2020, 14, 9050
2020	CNT/Mxene/silicone rubber-Kapton	Adv. Funct. Mater. 2020, 2004181
2021	Kapton-Nylon-Copper based alternating mode	Energy Environ. Sci. 2021, 14, 5395
2021	Electrode/Kapton-PET/Electrode rotational TENG	Nano Energy 2021, 87, 106170

Figure R1 The collection of the Kapton-based TENGs, indicating that Kapton is a frequently-used triboelectric material.

Comment 2:

In addition, the experimental data should be added for explaining the merits of 2D Bi₂Te₃ nanoplates. For example, Fig. 4e includes the power density of only nanomaterials. As illustrated in introduction, the figures should describe what the 2D TI materials improved compared to the conventional TENGs.

Response 2:

The authors would like to thank Reviewer #1's constructive comment on the performance comparison with conventional TENGs, and we agree with this viewpoint. To strengthen the significance of this work, we totally agree with the reviewer's suggestions that some systematical performance comparison among the measured TI-TENGs and conventional TENGs should be considered.

On the other hand, it is worth mentioning that the electrical characterizations of TENGs in most of the prior studies were not performed under standard conditions, for example, adopting uniform device area, operating force, and operating frequency. Thus, it is hard to conduct a fair comparison among them. Therefore, we select the power density of the TI-TENG as the typical merit for the performance comparison. **Figure R2** shows the power density comparison plot among conventional TENGs, emerging nanomaterial-TENGs, and TI-TENGs. In **Figure R2a**, TI-TENGs show distinct

advantages compared with emerging nanomaterial-TEGs. While in comparison with several conventional TENGs (**Figure R2b**), especially polymer-based TENGs, such as Kapton-based, PDMS-based, and Rubber-based, the performance advantage of TI-TEGs is eclipsed, further highlighting the direction of efforts for TI-TEGs. Although some improvements are still needed for TI-TEGs to compete with conventional TENGs, importantly, our research broadens both the existing triboelectric family and the application range of TI materials, which is significant for the fields of TENGs and TI materials.

Figure R2 Comparison plots of the output power density among previously reported TENGs based on (i) emerging nanomaterials, (ii) conventional materials, and TIs (Bi₂Te₃ and Bi₂Se₃).

Therefore, based on Reviewer #1's suggestions, to strengthen the significance of the TI-TEGs, the plot in **Figure R2b** has been supplemented in **Figure 4e** to make a comprehensive performance comparison. The relative description has been revised as "In **Figure 4e(i)**, the output power density of TI-TEGs could reach 180 mW m⁻², which could be competitive with emerging nanomaterials-based power sources, including MoS₂-based, graphene-based, and metal oxide frameworks-based counterparts. While the plot in **Figure 4e(ii)** eclipses the performance advantage of TI-TEGs compared with many conventional TENGs, especially polymer-based devices, potentially directing the improvement target for TI-TEGs." on **Page 11**. Relative references have been added as **Ref. 68–Ref. 73** in the new version of the manuscript. Thank the reviewer again for providing us with this constructive suggestion, which helps to improve the quality of our work.

Comment 3:

The quality and configuration of some figures should be improved for better understanding. So below papers are recommended to be referred to enhance the informativeness of figures.

- A. "Achieving high-resolution pressure mapping via flexible GaN/ZnO nanowire LEDs array by piezo-phototronic effect." *Nano Energy*, 58, 633-640 (2019)
- B. "Monolithic flexible vertical GaN light-emitting diodes for a transparent wireless brain optical stimulator." *Advanced Materials*, 30, 1800649 (2018)

Response 3:

The authors would like to thank Reviewer #1 for this suggestion and for kindly providing us with valuable references. Based on Reviewer #1's advice, **Figure 2**, **Figure 3**, and **Figure 4** have been thoroughly revised to improve the configuration and quality of this manuscript (as shown in **Figure R3-R5**). Also, two relative references have been added into the revised manuscript as **Ref. 18** (*Nano Energy* 2019, 58, 633-640) and **Ref. 19** (*Adv. Mater.* 2018, 30, 1800649) to support our revision.

Figure R3 The panel (c) has been revised to improve the configuration of the figure.

Figure R4 The panels (a), (b), and (c) have been revised to improve the configuration of the figure.

Figure R5 The panels (e) and (g) have been revised to improve the configuration of the figure and enrich the discussion.

Response to Reviewer #2:

Reviewer #2: In this manuscript, the authors investigated the feasibility to use low-cost Bi_2Te_3 -NP as the platform for energy-efficient triboelectric applications. Through detailed electric performance characterizations and various system demonstrations of the TI-based TENG, the authors did “fill the gap” and unfold a new avenue for the TI materials-based applications. In general, the work presented in this manuscript is quite original and inspiring in the TI research field, and it deserves the publication once the authors can address the following comments.

We thank Reviewer #2 for carefully reviewing our manuscript and providing us with insightful comments, which are very important for us to further improve the quality of the manuscript. According to the reviewer’s valuable comments, more experiments (e.g., **the demonstration of the Bi_2Se_3 -based TI-TENGs and the controlled experiment of the sapphire substrate-based TENG.**) have been conducted to support our discussion regarding the triboelectric properties of TI-TENGs and demonstrate the applicability of different TI nanomaterials in TENGs. In addition, relative revisions (including the **energy band structures of TI, the contributions of TIs to the triboelectric performance of TI-TENGs, and the details of the MBE-grown Bi_2Te_3**) have been made to strengthen the significance of this work. Each of the comments by Reviewer #2 is carefully considered and answered point-by-point. All corrections are highlighted in the revised manuscript.

Comment 1:

The biggest missing puzzle in this manuscript is what the role of the topological surface states in enhancing the performance of TI-based TENG. According to the authors’ argument, “the triboelectric charge densities that are strongly associated with the electron affinity difference between selected triboelectric materials and surface-contact modification engineering”. If this is true, it seems that the only key factor of TI for TENG application is the relatively large surface potential (Fig. 3b). Under such circumstances, I do not recommend not to overstate the topological surface properties in this manuscript. After all, Bi_2Te_3 has been widely used long before it was “re-named” as topological insulators.

Response 1:

We thank Reviewer #2 for the advice regarding the role of the topological surface states and agree that the topological surface properties should not be overstated. We apologize for causing the confusion that the topological surface properties (e.g., the spin-momentum locking) of Bi_2Te_3 -NP film are the key leading to the triboelectric performance of our TI-TENGs. Prior studies regarding the electrification effect of TENGs (please refer to *Energy Environ. Sci.* 2015, 8, 2250; *Nano Lett.* 2013, 13, 2771; *Nat. Commun.* 2019, 10, 1427) suggest that the output performance of TENGs is strongly related to the transferred charge densities, which can be improved by proper material selection and contact surface engineering. Accordingly, the main points that contribute to the triboelectric performance of our devices could ascribe to three aspects:

- (1) The **surface potential difference** between two triboelectric materials we choose (e.g., Bi_2Te_3 and Kapton) is relatively large;
- (2) The solution-based Bi_2Te_3 film that consists of numerous nanoplates effectively improves the **contact behaviors** between triboelectric layers;
- (3) Triboelectrification is dominated by the **surface charge transfer** between two materials and has little dependence on their bulk property. Hence, the TI materials with **metallic surface states** and insulating bulk are considered significant candidates for triboelectric experiments.

Therefore, we believe that the origins of the performance enhancement of TI-TENGs should be attributed to the synergistic contributions from these three points. In addition, to strengthen our argument about the TI materials' contributions to the electrical performance of TENGs, we further examine the triboelectric properties of another typical TI material— Bi_2Se_3 via material synthesis, material characterization, KPFM analysis, device fabrication, and electrical measurement. As shown in **Figure R6a**, the synthesized Bi_2Se_3 is also hexagon nanoplate, showing similar surface morphology as Bi_2Te_3 . **Figure R6b** shows the KPFM analysis of the surface potential of Kapton before and after contact with Bi_2Se_3 -NP film, in which the potential difference between the probe tip and Kapton increases after contact with Bi_2Se_3 . This phenomenon indicates that Bi_2Se_3 possesses negative charge electrification compared with Kapton. Also, the different surface potential change of Bi_2Se_3 and Bi_2Te_3 ($\Delta E = 1.14$ eV) primarily hints at their different charge electrification ability in the triboelectric series.

Furthermore, we examine the electrical performance of the Bi_2Se_3 -NP film-based TENG by taking Kapton as the counterpart triboelectric layer. The voltage outputs and transfer charges of the Bi_2Se_3 TI-TENG in **Figure R6c-6d** reveal a similar triboelectric performance as Bi_2Te_3 TI-TENG under the same measurement conditions. These results indicate that the similar surface morphology and different surface potential lead to the similar triboelectric performance of Bi_2Se_3 and Bi_2Te_3 -based TENGs. Correspondingly, we could qualitatively deduce that besides the surface potential and surface morphology, other factors, such as the surface conducting property, also contributes to the triboelectric properties of the TI-TENGs. Thus, we believe that the investigation of the Bi_2Se_3 nanomaterial and Bi_2Se_3 -NP film-based TENG is highly consistent with the origins we proposed above, which primarily supports our discussion.

Figure R6 The investigation of the Bi_2Se_3 and Bi_2Se_3 -NP TI-TENG. (a) SEM image of the hexagonal Bi_2Se_3 nanoplates (Bi_2Se_3 NP). (b) Surface potential variation of Kapton film (i) before and (ii) after contact with Bi_2Se_3 -NP film. (c) and (d) show the measured output voltage and transferred charges of the Bi_2Se_3 -NP TI-TENGs with Kapton as the counter triboelectric layer.

Therefore, the authors appreciate this professional comment from Reviewer #2 and agree to rigorously revise the relative description to avoid overstating the topological surface states properties of TIs. Based on the reviewer’s suggestions, to make a rigorous description, relative revisions include:

“Albeit never involving the field of nanogenerators, TIs’ unique surface conducting property makes them an ideal candidate material for TENGs since the triboelectrification is strongly dominated by the surface charge transfer process between tribolayers.” on **Page 4**;

“Both the larger surface potential difference and the conductive surface states of the nanofilm synergistically improve the charge transfer behavior between the selected triboelectric media, rendering a TI-based TENG (TI-TENG) with considerable output performance.” in the **Abstract**;

“The surface conducting property, improved contact behavior, and larger surface potential difference with Kapton endows Bi_2Te_3 -based TI-TENGs with enhanced triboelectric charge transfer ability and considerable output power performance.” on **Page 4**.

“Such electrification effect of TIs was further validated by examining the triboelectric behavior of another typical TI material— Bi_2Se_3 . Under the same measurement conditions, Bi_2Se_3 -NP film-based

TENG exhibits similar triboelectric performance with Bi_2Te_3 TI-TENG (**Figure S7**). The systematical investigation of the surface morphology and surface potential variation further indicates that the triboelectric performance of TI-TENGs originates from the synergistic contributions from surface potential, contact behavior, and the conducting property of TI materials.” on **Page 9** to support our discussion.

The investigation of the Bi_2Se_3 -NP-based TENG in **Figure R6** has been supplemented as **Figure S7** to support our discussion. Also, relative references including **Ref. 47** (*Energy Environ. Sci.* 2015, 8, 2250) and **Ref. 48** (*Mater. Today* 2017, 20, 74) are added to support our conclusion. Thanks very much for the reviewer’s constructive comment, which is valuable for improving the quality of the manuscript. We hope that our effort and explanation can provide an insightful understanding of the TI materials’ role in the electrical performance of TI-TENGs to Reviewer #2.

Comment 2:

Likewise, the authors should clearly elaborate the enhancement of TENG performance of Bi_2Te_3 -NP is due to the increased surface-to-bulk volume or topological surface state features (e.g., higher surface mobility, spin-to-momentum locking, and suppressed back-scattering). It should be noted that the surface state contribution is normally negligible at room temperature owing to the dominant thermal-activated bulk condition.

Response 2:

The authors would like to thank Reviewer #2 for the comments regarding the origins of the performance enhancement of TI-TENGs. As the reviewer stated, the transport inside TIs is dominated by the thermally activated bulk carriers at room temperature. However, contact electrification relies almost only on the surface charge transfer between two triboelectric materials, which means that triboelectrification will be governed by the surface properties of the triboelectric materials, and the bulk properties of the material will become negligible. Considering the conductive surface property of TI materials, we believe that the **surface charge transfer** has a specific contribution to the triboelectric property of TI-TENGs.

In addition, as we mentioned in question #1, both the surface potential difference and the contact behavior between two triboelectric layers are essential factors to the charge transfer process. Thus, we can preliminarily conclude that the main origins for the electrical performance of TI-TENG include: **(1)** a large surface potential difference between Bi_2Te_3 -NP film and Kapton, **(2)** a good contact behavior benefiting from the high surface-volume ratio and ultrathin thickness of the Bi_2Te_3 -NP film, and **(3)** the conducting surface properties of TIs.

On the other hand, the scope of this work mainly focuses on exploiting the potential of TI nanomaterials in the TENG field and characterizing the triboelectric ranking of Bi_2Te_3 -NP films in the triboelectric series. Whether the spin-momentum locking and high mobility of the surface states play a role in enhancing the electrical output of TENGs is still unclear. More detailed transport and

scanning probe microscopy studies at low temperatures to investigate the actual influence of the surface states' heliacal property and high mobility will be a critical part of our future works.

Therefore, based on Reviewer #2's suggestions, to elaborate the reasons for the performance enhancement of the TI-TENGs, the relative description has been revised, including:

- (1) The sentence "Energetic electronic states on the surface of nanofilm improve the charge transfer efficiency, affording a TI-based TENG (TI-TENG) with considerable output performance." has been revised as "Both the larger surface potential difference and the conductive surface states of the nanofilm synergistically improve the charge transfer behavior between the selected triboelectric media, rendering a TI-based TENG (TI-TENG) with considerable output performance." in the **Abstract**.
- (2) The sentence "The unique surface electronic structure of TI film endows TI-TENG with enhanced triboelectric charge density and outstanding output power performance." has been revised as "The surface conducting property, improved contact behavior, and larger surface potential difference with Kapton endows Bi₂Te₃-based TI-TENGs with enhanced triboelectric charge transfer ability and considerable output power performance." on **Page 4**.
- (3) The sentences "Albeit never involving the field of nanogenerators, these attractive characteristics of TIs are believed to ideally meet the benchmarks of triboelectric materials. Concretely, numerous energetic states on the surface enhance the surface charge densities, and the assembled-nanoplate film (Bi₂Te₃-NP film) contributes to surface charge trapping/detrapping. Furthermore, Bi₂Te₃ NPs are expected to have tunable dielectric properties due to their conducting surface states, which extend the degree of freedom to adjust the output performance of TENGs." have been revised as "Albeit never involving the field of nanogenerators, TIs' unique surface conducting property makes them an ideal candidate material for TENGs since the triboelectrification is strongly dominated by the surface charge transfer process between tribolayers." on **Page 4**.
- (4) The sentences "Notably, the electrical performance of the Bi₂Te₃-NP film in Figure 2d was slightly better than that of MBE-based Bi₂Te₃, which is attributed to the increased surface electronic states in the assembled-nanoplate film as a result of abundant defective edges. In addition, larger spacing between nanoplates in the Bi₂Te₃-NP film could mitigate the negative effects of layer-layer interactions on the electric properties. Thus, it is believed that the unique electronic states on the surface of TI materials, particularly in nanostructures, are conducive to promoting the local conductivity and contribute to the pronounced triboelectric charging behaviors of TI-TENGs." have been revised as "Notably, the slight performance difference between these two devices could originate from multiple factors, such as the different effective contact areas or interlayer interactions." on **Page 7**.

- (5) The sentence “In summary, a topological insulator—a new triboelectric medium with unique surface electronic states—was introduced into the triboelectric series and the field of triboelectric energy.” has been revised as “**In summary, a topological insulator—a new triboelectric medium with favorable surface charge properties—was introduced into the triboelectric series and the field of triboelectric energy.**” on **Page 14**.
- (6) The sentence “Benefiting from the enhanced surface electronic density, the Kapton-Bi₂Te₃ film paired TI-TENG exhibited considerable output power, reproducible energy-harvesting capabilities, and the ability to drive portable electronics.” has been revised as “**Benefiting from the enhanced surface charge transfer process, the Kapton-Bi₂Te₃ film paired TI-TENG exhibited considerable output power, reliable energy-harvesting capabilities, and the ability to drive portable electronics.**” in the part of **Conclusion**.

Also, relative references have been added as **Ref. 47** (*Energy Environ. Sci.* 2015, 8, 2250), **Ref. 48** (*Mater. Today* 2017, 20, 74), and **Ref. 57** (*Nano Lett.* 2013, 13, 2771) to support our discussion. The authors appreciate the reviewer very much for putting up forward this constructive comment. We hope that our effort could offer an insightful understanding to Reviewer #2.

Comment 3:

In Fig. 2, the authors also compared the TI-NP with MBE-grown TI film. However, there is no information regarding the MBE-based TI control samples, e.g., film thickness, area, substrate, carrier density etc.

Response 3:

The authors would like to thank Reviewer #2 for pointing out this question and agree with the reviewer that detailed information of the MBE-based TI sample should be provided. In our work, to validate the triboelectric properties of TI materials, we investigated the output behavior of the MBE-based TI TENG. To make a fair comparison, both the samples and measurements are made as similar as possible to the solution-based TI-TENG.

Concretely, the fabricated MBE TI-TENG consists of PET/ITO, Kapton, MBE-grown Bi₂Te₃ film on the sapphire substrate, and ITO from the bottom to the top. The thickness and area of the MBE-grown Bi₂Te₃ film are around 5 nm and 0.7 cm×0.7 cm, identical to the solution-based TI TENG for comparison. The measured carrier density of the single-crystal film is around $4 \times 10^{13} \text{ cm}^{-2}$ at room temperature. Note that sapphire serves as the substrate for the Bi₂Te₃ film synthesis. To exclude the impact of sapphire dielectric on the triboelectric performance of MBE-based TENG, we conducted another controlled experiment (sapphire-Kapton-based TENG) to confirm the fairness of the comparison. As shown in **Figure R7**, the output performance of the sapphire-Kapton-based TENG reveals that the sapphire substrate’s contribution to the MBE-based TI-TENG is negligible, indicating a fair comparison between MBE-based and solution-based TI-TENGs.

Figure R7 Controlled experiment of the sapphire-Kapton-based TENG. (a) The measured voltage signal, (b) transferred charges, and (c) measured current signal of the sapphire-Kapton-based TENG, indicating the negligible contributions from the sapphire substrate to the performance of the MBE-based TI-TENG.

Therefore, the authors would like to thank Reviewer #2 again for proposing this question. Based on Reviewer #2's suggestion, to make a clear presentation, the demonstration of the sapphire-Kapton-based TENG in **Figure R7** is added as **Figure S5** to exclude the contribution of the sapphire substrate. The relative description has been revised as “For a fair comparison, the measurement conditions for MBE TI-TENG were made as similar as possible to the solution-based TI TENG.” on **Page 7**. The sentence “The film thickness and area of the MBE-grown Bi_2Te_3 are 5 nm and 0.5 cm^2 , which are identical with the solution-based TI-TENG for a fair comparison.” has been added in the caption of **Figure S4**. We hope that our effort can offer clear information of the MBE-grown TI film to Reviewer #2.

Comment 4:

Line 74 – Bi_2Te_3 is conventionally categorized as a 3D TI instead of 2D TIs (e.g., CdTe/HgTe, InAs/GaSb heterostructures).

Response 4:

The authors would like to thank Reviewer #2 for kindly pointing out this issue about the classification of the Bi_2Te_3 , and we agree with the reviewer's viewpoint. According to the quantum spin Hall effect, conventional two-dimensional topological insulators include CdTe/HgTe, InAs/GaSb, etc. While Bi_2Te_3 , Bi_2Se_3 , and Sb_2Te_3 should be categorized as 3D topological insulators whose surface states consist of a single Dirac cone at the point. Therefore, based on the reviewer's suggestion, to make a correct presentation, relative revision has been made as “Bismuth telluride (Bi_2Te_3), a compound that has been extensively studied for its thermoelectric properties, was found to be a 3D TI with a quintuple-layered structure and gained significant attention in recent years.” on **Page 3**. The authors thank Reviewer #2 again for carefully reviewing and correcting our manuscript.

Comment 5:

The band structure of Bi_2Te_3 in Fig. 2b is not correct.

Response 5:

The authors would like to appreciate Reviewer #2's kind suggestions about the band structure, and we apologize for the carelessness. Based on the reviewer's advice, to make a clear presentation, the band structure in Figure 2b has been corrected as shown in **Figure R8** and has been renewed in the new version of the manuscript. Many thanks to Reviewer #2 again for kindly correcting this point.

Figure R8 The sketch of the band structure of Bi_2Te_3 with a typical bulk bandgap (bulk conductive band/BCB, bulk valence band/BVB, black lines) and unique conductive surface state band (yellow line, SSB).

Comment 6:

The authors claimed that “the electrical performance of the Bi_2Te_3 -NP film in Figure 2d was slightly better than that of MBE-based Bi_2Te_3 , which is attributed to the increased surface electronic states in the assembled-nanoplate film as a result of abundant defective edges.” If this is true, it seems that the high performance of TI-based TENG is more related to the surface defects other than the original topological surface states.

Response 6:

The authors would like to thank Reviewer #2's professional comments on the comparison of solution-based and MBE-based TENGs. We apologize for causing the misleading that the surface defects of the Bi_2Te_3 -NP film determine the triboelectric performance of our TI-TENGs. As we proposed above, the triboelectric performance of the Bi_2Te_3 -NP films should be attributed to the synergistic effects, including the surface potential difference of two triboelectric layers, good contact behavior, and surface conducting property of TI film. In terms of two different fabrication methods, Figure 2d revealed a slight performance difference between the two devices. Such a small difference could originate from several factors, such as the larger effective surface area of solution-based TI film (abundant defective edges) than MBE-based TI film and the better contact behavior of solution-based TI-TENG due to the flexible PET substrate. However, it is worth mentioning that **the performance difference between these two devices is not considerable**. This indicates that the contributions from defective edges of nanomaterials or the flexible substrates to the triboelectric performance cannot compete with the surface potential difference between trilayers, which coincides with our previous discussion.

Therefore, based on the reviewer's suggestions, to make a clear presentation and avoid misleading, relative description has been revised as “Notably, the slight performance difference between these

two devices could originate from multiple factors, such as the different effective contact areas or interlayer interactions.” on **Page 7**. Thank Reviewer #2 again for giving this professional suggestion, and we hope that our explanation can offer a clear understanding to Reviewer #2.

Comment 7:

Besides, I notice that the band diagrams shown in Fig. 3c do not include the TI surface states.

Response 7:

The authors would like to thank Reviewer #2’s professional suggestion about the band diagrams of Bi_2Te_3 , and we agree with the reviewer’s viewpoint that the TI surface states are not included in our previous version. In the prior version of the manuscript, Figure 3c shows the brief band diagrams of two triboelectric layers and the charge transfer behaviors after connecting. To make these two charge transfer processes in Figure s3c(i) and 3c(ii) more straightforward, the surface states of TI materials were not emphasized. However, these simplified energy band diagrams might lead to some misleading due to the missing surface states. We apologize to Reviewer #2 for the misleading information and agree that the TI surface states should be added to the band diagram to make the figures more rigorous. Therefore, based on the reviewer’s suggestions, to make a clear presentation, the surface states of Bi_2Te_3 are supplemented in the band diagrams (as shown in **Figure R9**). This revision has been renewed (**Figure 3c**) in the revised manuscript. Thanks a lot for Reviewer #2’s valuable comments, and we hope the modification can offer a clear understanding to Reviewer #2.

Figure R9 Energy band diagram for Bi_2Te_3 -NP film when contacting (a) positively electrified materials and (b) negatively electrified materials. The crossed symbols indicate the conductive surface state of TI material.

Response to Reviewer #3:

Reviewer #3: This paper describes an experimental work using a topological insulator material, Bi_2Te_3 , to construct triboelectric nanogenerators. As authors claim, triboelectric nanogenerators are becoming a good choice in powering self-powered devices, sensors, etc. Although there are innumerable works and substantial development of the practical approaches in making up these devices, since the devices are made up mostly of organic polymers (at least on one electrode), plasticity is an issue. The new class of materials the authors suggest and the example they show in their report seems to solve some of the problems in current TENG technology.

The authors clearly show the preparation of the material and the manufacturing of the devices with the new material. They provide two ways of materials preparation (solution-based and MBE-based), both of which work to deliver the targeted material - with slight differences in the device. Also, working with conventional materials on the other electrode, they built a set of TENG devices and have shown that the new material added to the device increases the charge transfer abilities considerably. They also described the charge generation mechanism by showing that the charge generation is due to an electron transfer between the surfaces – the direction of which is dependent on the ranking of the materials with respect to each other in the triboelectric series. This mechanism is supported well with the KPFM potential maps.

In addition to all these, the display of applications with the new material shows versatility.

Overall, the claims are supported well and the methodology used meets the standards in the field. I suggest only minor additions to the manuscript.

We thank Reviewer #3 very much for carefully reviewing our manuscript and providing us with constructive comments that are important for further improving the quality of the manuscript. According to the valuable suggestions, more explorative experiments (**1. TI-TENG based on identical triboelectric material of Bi_2Te_3 -NP films; 2. The endurance test of the prolonged operations over 20000 cycles; 3. The morphological evolution of the triboelectric materials after a prolonged operation.**) have been performed to investigate more triboelectric properties of the TI-TENGs and examine their excellent durability for practice uses. The authors carefully revised the manuscript based on Reviewer #3's professional comments. To make a clear presentation, each comment is answered point-by-point, and the related corrections are highlighted in the revised version of the manuscript.

Comment 1:

It could have been interesting to show 'identical material' contact with the Bi_2Te_3 material on both electrodes. I guess there would be no charging, which I would expect due to the electron transfer mechanism. In cases with organic polymers, this is never the case.

Response 1:

The authors would like to thank Reviewer #3's constructive suggestions about the

contact-electrification (CE) of two identical triboelectric materials. In general, a triboelectric nanogenerator consists of two different triboelectric dielectric materials. Recently, Xu *et al.* have demonstrated the triboelectrification between two identical materials such as polytetrafluoroethylene (PTFE), fluorinated ethylene propylene (FEP), and Kapton (please refer to ACS Nano 2019, 13, 2034-2041). Based on the contact-separation mode, it reports that two pieces of identical materials can result in electrostatic charges, although the charge density is relatively low. About the proposed charge transfer model, it states that CE between identical materials shows no apparent tendency of the charge transfer direction for two flat films. While for two curvature films, CE between identical materials shows the curvature-dependent charge transfer process because of the curvature-induced energy shifts of the surface states.

Accordingly, based on the reviewer's suggestion, to further investigate the triboelectric behavior of two identical Bi_2Te_3 -NP films, we conduct a supplementary experiment by fabricating a contact-separation Bi_2Te_3 - Bi_2Te_3 TENG and characterizing the electrical performance. As shown in **Figure R10**, the collected voltage signal indicates that several electrostatic charges are generated by two Bi_2Te_3 films, which could be attributed to the initial residual charges on the surface of Bi_2Te_3 films or the curvature difference between the two films. Besides, Bi_2Te_3 - Bi_2Te_3 TENG delivers a low voltage with evident noise, suggesting a small amount of the generated electrostatic charges. This is highly consistent with the reported results (please refer to ACS Nano 2019, 13, 2034-2041), which measured a very low output voltage (<1 V) even at a high contact force of 30 N.

Thus, we can preliminarily deduce that CE exists between two identical Bi_2Te_3 films, while the triboelectric performance cannot be competitive with a couple of asymmetry triboelectric materials. We hope that our efforts can provide a clear explanation for Reviewer #3. Besides, we also would like to thank Reviewer #3 again for your constructive suggestions that inspire more interesting thoughts for TENG research.

Figure R10 The demonstration of the identical triboelectric materials-based TENG. (a) The measured voltage signal, (b) transferred charges, and (c) measured current signal of the Bi_2Te_3 - Bi_2Te_3 -based TENG.

Comment 2:

The mechanism and in prolonged operation in real-life cases, there can be material transfer from the other electrode material (polymer) to the topological insulator, which I think is mechanically harder. Then the contact may be a polymer(deposited on Bi_2Te_3) – polymer rather than the pristine material contact. Can authors comment on this? Is it possible to use no polymers with the new material TENGs?

Response 2:

The authors would like to thank Reviewer #3 for this professional comment on the durability of the triboelectric materials. Based on Reviewer #3 suggestions, we conducted a prolonged endurance test to examine the durability of TENGs that consist of Bi_2Te_3 -NP film and Kapton. As shown in **Figure R11**, similar to the result in Figure 4g, the collected voltage signals over 20000 continuous cycles remain steady without evident fluctuations, revealing that the triboelectric material is exceptionally stable and durable.

Figure R11 The demonstration of the prolonged operation of the TI-TENG over 20000 cycles with a negligible performance degradation indicates the excellent durability of the device.

On the other hand, as to the material transfer in prolonged operation in real-life cases, we agree with Reviewer #3's viewpoint because material transfer cannot be avoided entirely. To examine the material transfer phenomenon between two triboelectric layers, we further characterize the surface morphology change of the Kapton by SEM analysis. **Figure R12** shows the SEM images of the Kapton films before (Bef-Kapton) and after (Aft-Kapton) operating 20000 cycles. Besides a few discrete Bi_2Te_3 NP or impurities on the Aft-Kapton's surface (marked by the arrows in **Figure R12(b)**), no distinguished difference between Bef-Kapton and Aft-Kapton can be found. This phenomenon indicates that a mild material transfer from Bi_2Te_3 -NP film to Kapton occurs after a prolonged operation, instead of from Kapton to Bi_2Te_3 -NP film considering the commercial Kapton polymer film consists of closely linked chains. It believes that such a mild material transfer could be resolved by a thermal annealing treatment of the Bi_2Te_3 -NP film. Also, the steady output signals over 20000 cycles reveal that such a gentle material transfer between two triboelectric layers does not

significantly impact the electrical performance of the TENG devices and could be neglected.

Figure R12 The morphology comparison of the Kapton triboelectric layer after prolonged operation. SEM images of the Kapton film (a) before and (b) after 20000 cycles. The arrows in (b) indicate the few impurities such as Bi₂Te₃ NP on the surface of Kapton after prolonged operation, which lead to negligible impacts on the electrical performance of the TI-TENGs.

In addition, based on the reviewer’s suggestions, we also fabricate a no-polymer TENG to examine the applicability of TI material in non-polymer TENGs. Taking the glass as an example, the results in **Figure R13** show that the Bi₂Te₃-glass-based TENG delivers repeatable outputs. It is believed that such output performance can be improved by elaborately optimizing the contact behaviors and matching the surface potential of the triboelectric counterparts, revealing the applicability of TI materials in no-polymer TI-TENGs.

Figure R13 The demonstration of the non-polymer TI-TENG. (a) The voltage output, (b) transferred charge, and (c) the current signal of the Bi₂Te₃-glass-based TENG, which indicate the applicability of the TIs in non-polymer TI-TENGs.

Therefore, based on Reviewer #3’s suggestions, to further emphasize the durability of the TI-TENGs, the results of the prolonged operation over 20000 cycles and SEM images are added in the revised manuscript as **Figure 4g** and **Figure S11**. The relative description “Note that the negligible fluctuation over 20000 cycles could be attributed to the gentle material transfer between two triboelectric layers or the noise signal from the external circuits, which is consistent with the surface

morphology variation of Kapton (Figure S11).” has been added on Page 11 to strengthen our discussion. We hope that our effort and explanation can offer an insightful understanding of the durability of the TI-TENGs and the applicability of the non-polymer TI-TENGs to Reviewer #3.

Comment 3:

It was mentioned in the text that ‘The tunability of the output—via both the dose of Bi₂Te₃ NPs and the size of the Bi₂Te₃-NP films—enable good scalability of TI-TENGs for practical uses.’

Was there some correlation between the surface area (size) of the nanoplates and the performance metrics of the device /efficiency of charge transfer?

Response 3:

The authors would like to thank Reviewer #3 for raising the valuable comment on the correlation between the surface area of triboelectric materials and device performance. In our work, the size of the synthesized Bi₂Te₃ nanoplates is around 500-550 nm (the distance between two parallel sides), which is fixed. The changeable sizes are the surface areas of the Bi₂Te₃-NP film and Kapton triboelectric layers. Based on the reviewer’s comment, we discuss the correlation between the surface area of the triboelectric materials and performance in two aspects: **(1)** surface area of the Bi₂Te₃ film **(film)**, and **(2)** size of the Bi₂Te₃ nanoplate **(sample)**, respectively.

On the one hand, in terms of the surface area of film-dependent triboelectric performance, results in Figure 4b (main text) indicate that the measured voltage increases from 55 V to 165 V with the contact area of the triboelectric layers increasing from 1 cm² to 25 cm². We further extract the plot of voltage versus surface area to examine the corresponding relationship. As shown in Figure R14, the voltage signal is near-linear related to the contact area, and the change rate, which is the slope of the fitted line, is around 4 V/cm². This result indicates that the fabricated TI-TENG is scalable for practical uses and highly consistent with previous conclusions (please refer to *Nano Lett.* 2013, 13, 2771; *Energy Environ. Sci.* 2015, 8, 2250), which stated that the triboelectric-induced transfer charges are proportional to the contact area between triboelectric layers.

On the other hand, in terms of the sample size, we agree that it theoretically has some impact on the triboelectric performance of TI-TENGs because different sample sizes lead to the variation of the film areas. However, we consider that the sample size’s impact on the triboelectric performance is insignificant based on the following reasons. **(1)** As shown in Figure 2d (main text), the comparison experiment of Bi₂Te₃-NP-based and Bi₂Te₃-MBE-based TENGs revealed that the performance difference originated from two different films is small. **(2) Different sample sizes-induced surface area variation is extremely low**, which is lower than the area difference between Bi₂Te₃-NP-based and Bi₂Te₃-MBE-based films. Thus we can reasonably deduce that its impact on the triboelectric performance should be negligible.

Moreover, the scope of our work mainly focuses on exploiting the potential of TI nanomaterials in the new field of TENGs and characterizing its triboelectric ranking in the triboelectric series. As to

the specific correlation between the sample size and the triboelectric performance, we believe that more rigorous material fabrication and characterizations are still needed, which provides a worthy topic for our future research.

Figure R14 The output voltage of the Bi₂Te₃-NP TENG depends on the contact area varying from 1 to 25 cm². The fitting line (red line) indicates the linear relationship between the voltage signal and the device area.

Therefore, based on Reviewer #3's suggestion, to make a clear presentation, the sentences have been revised as "In Figure 4b, the output voltage shows a proportional increase with the increasing area of the TI film, which is consistent with the proportional relationship between the total amount of the transferred charges and the contact area. The fitting result is provided in Figure S10. The tunability of the output—via both the dose of Bi₂Te₃ NPs and the size of the Bi₂Te₃-NP films—suggests the high practicality of TI-TENGs." on Page 10. A sentence has been revised as "Both the larger surface potential difference and the conductive surface states of the nanofilm synergistically improve the charge transfer behavior between the selected triboelectric media, rendering a TI-based TENG (TI-TENG) with considerable output performance." in the **Abstract**. Also, the fitting plot in Figure R14 is supplemented as Figure S10, and two relative references are added as Ref. 47 (*Energy Environ. Sci.* 2015, 8, 2250) and Ref. 57 (*Nano Lett.* 2013, 13, 2771) to support our conclusions.

The authors would like to appreciate Reviewer #3's comment and hope our explanation can offer an insightful understanding to Reviewer #3.

Comment 4:

Does the synthesis provide quintuple layers all the time? For the unfamiliar reader, it is better to describe this part a bit.

Response 4:

The authors would like to thank Reviewer #3's valuable comment about the material properties. In our work, the XRD pattern in Figure 1f revealed that the synthesized Bi₂Te₃ possesses the

rhombohedral crystal structure, which is consistent with the experimental and theoretical results in previous studies. As shown in **Figure R15**, the building block of Bi_2Te_3 is one quintuple layer (QL) (please refer to *Nat. Mater.* 2009, 5, 438; *Adv. Mater.* 2010, 22, 4002). It consists of five-atom layers with a stacking sequence of Te(1)-Bi-Te(2)-Bi-Te(1) along with the z-direction and is terminated by one Te(1) layer at both ends. Each QL cell is coupled with two adjacent QL cells by van der Waals force, which is much weaker than the interaction between two atomic layers within one QL cell. This hints that the Te-terminated structure as the cleaved surfaces of the Bi_2Te_3 is energetically favorable. Thus, the well-synthesized Bi_2Te_3 should be QL-structured, especially the MBE-grown samples.

On the other hand, in other synthesis methods (such as the hydrothermal and chemical vapor deposition methods), various structural or compositional defects (e.g., vacancies, dislocations, and defective edges) are difficult to avoid during the material synthesis. Although abundant defects might lead to specific localized structural variations at the top and bottom surfaces of Bi_2Te_3 nanomaterials, the QL structure can be well-maintained as the building block inside (please refer to *Nano Lett.* 2010, 10, 2245).

Therefore, based on Reviewer #3's suggestions, to make a clear presentation, related description of the material structure has been revised as "The schematic diagrams of the crystal structure and energy band of Bi_2Te_3 in Figure 2b(i) highlight its building block—quintuple layer (QL) and the unique surface conductive features. Each QL cell consists of five layers, which are stacked by a sequence of Te(1)-Bi-Te(2)-Bi-Te(1) along the z-direction and terminated by a Te(1) layer at both ends. Compared with the strong interaction within each QL cell, the van der Waals force between adjacent QL cells is much weaker, leading to the preferential cleave surface of Te atomic layer." on **Page 6**. The schematic of the QL cell in Figure 2b has been revised as **Figure R15**. Relative reference is also added as **Ref. 54** (*Nat. Mater.* 2009, 5, 438). We hope that our explanation can offer an insightful understanding to Reviewer #3.

Figure R15 The crystal structure of the quintuple layer (QL) Bi_2Te_3 , which is stacked by a sequence of Te(1)-Bi-Te(2)-Bi-Te(1) along the z-direction.

Thank you in advance for your time and kind consideration of our manuscript.

Sincerely Yours,

Prof. Dr. Ying-Chih Lai

Chief Editor of Journal of Taiwan Vacuum Society

Department of Materials Science and Engineering,

Research Center for Sustainable Energy and Nanotechnology,

Innovation and Development Center of Sustainable Agriculture, National Chung Hsing University

National Chung Hsing University

No 250, Guoguang Rd., Taichung City 402, Taiwan

Tel: +886-4-22840500-300

Research group website: <https://lai423.wixsite.com/advdevice>

E-mail: yclai@nchu.edu.tw

REVIEWERS' COMMENTS

Reviewer #1 (Remarks to the Author):

Regarding the theoretical explanation and data supplement, the author clearly addressed the reviewers' comments in the revised manuscript. This work contributes to the advances in the TENG by using the 2D TI materials, and thus deserves a recommendation of acceptance in Nature Communications.

Reviewer #2 (Remarks to the Author):

In this revised manuscript, the authors have made great efforts to address all comments raised by the reviewers. Now I am quite satisfied by the updated version and recommend the publication of this exciting work on Nature Communications.

Reviewer #3 (Remarks to the Author):

In the revised version, all the reviewer comments are addressed and questions are answered in detail. The present version contains convincing and comprehensive data. I thank the authors for providing diligent explanations to my questions.